# HTR6 and SSTR3 ciliary targeting relies on both IC3 loops and C-terminal tails

Pablo Barbeito[1,2,3] ⓘ, Yuki Tachibana[4], Raquel Martin-Morales[1,2,3] ⓘ, Paula Moreno[1,2], Kirk Mykytyn[5,6], Tetsuo Kobayashi[4] ⓘ, Francesc R Garcia-Gonzalo[1,2,3] ⓘ

**G protein-coupled receptors (GPCRs) are the most common pharmacological target in human clinical practice. To perform their functions, many GPCRs must accumulate inside primary cilia, microtubule-based plasma membrane protrusions working as cellular antennae. Nevertheless, the molecular mechanisms underlying GPCR ciliary targeting remain poorly understood. Serotonin receptor 6 (HTR6) and somatostatin receptor 3 (SSTR3) are two brain-enriched ciliary GPCRs involved in cognition and pathologies such as Alzheimer's disease and cancer. Although the third intracellular loops (IC3) of HTR6 and SSTR3 suffice to target non-ciliary GPCRs to cilia, these IC3s are dispensable for ciliary targeting of HTR6 and SSTR3 themselves, suggesting these GPCRs contain additional ciliary targeting sequences (CTSs). Herein, we discover and characterize novel CTSs in HTR6 and SSTR3 C-terminal tails (CT). These CT-CTSs (CTS2) act redundantly with IC3-CTSs (CTS1), each being sufficient for ciliary targeting. In HTR6, RKQ and LPG motifs are critical for CTS1 and CTS2 function, respectively, whereas in SSTR3 these roles are mostly fulfilled by AP[AS]CQ motifs in IC3 and juxtamembrane residues in CT. Furthermore, we shed light on how these CTSs promote ciliary targeting by modulating binding to ciliary trafficking adapters TULP3 and RABL2.**

## Introduction

Primary cilia are microtubule-based cellular antennae that sense extracellular stimuli in a cell type-specific manner. To do this, each cell type must target specific receptors and signal transducers to its cilia. For instance, limb and neural progenitors in vertebrate embryos must target Patched, Smoothened (SMO), and other Hedgehog (Hh) pathway components to their primary cilia if they are to properly respond to Sonic Hedgehog (Shh), an essential morphogen (1). Likewise, kidney epithelial cells must traffic the mechanosensitive polycystin channel to their ciliary membrane to sense urine flow (2). Failure to do so leads to polycystic kidney disease, the most common

of the ciliopathies, a diverse group of human diseases resulting from cilia malfunctions (3). In fibroblasts and mesenchymal stem cells, cilia accumulate PDGFRα and IGF1R, two growth factor receptors affecting directional cell migration and adipogenesis, respectively (4). Similarly, many neurons throughout the central nervous system (CNS) have primary cilia whose signaling relies on G protein–coupled receptors (GPCRs) and their effectors (5, 6).

GPCRs are the most common drug target in the human clinic (7). The human genome encodes nearly 1,000 GPCRs, most of them olfactory receptors (ORs). ORs function in the cilia of olfactory sensory neurons, from nematodes to man. Vision also relies on ciliary GPCR signaling, requiring accumulation of Opsins and their effectors inside retinal photoreceptor cilia. Besides connecting us to the outside world, ciliary GPCRs also sense a myriad of endogenous signals. Some of these GPCRs work outside the nervous system, most in epithelial cells (renal, thyroid, bile duct, airways, endothelium, etc.) but others in mesenchymal stem cells (FFAR4) or fibroblasts, where SMO, GPR161, and GPR175 control Hh pathway output. Still, even after discounting visual and olfactory GPCRs, most known ciliary GPCRs function in neurons, detecting either neuropeptides (e.g., melanocortin, kisspeptin, galanin, melanin-concentrating hormone, and somatostatin) or neuroactive amines such as serotonin and dopamine (5, 6, 8).

For olfactory and visual GPCRs, ciliary localization has long been known to be essential for function (5, 6). This, too, is becoming increasingly clear for other ciliary GPCRs. For instance, some obesity-causing mutations in melanocortin receptor 4 (MC4R) act by preventing its ciliary targeting, thereby perturbing adenylyl cyclase 3 (AC3)-dependent cAMP signaling in hypothalamic cilia (9). Somatostatin receptor 3 (SSTR3) is expressed in cilia throughout the brain, where it colocalizes with AC3. Remarkably, mice lacking SSTR3, AC3, or cilia in the hippocampus all display very similar learning and memory defects, suggesting that ciliary targeting of SSTR3 and AC3 are essential for these processes (5). SSTR3 is also expressed outside the brain, mostly in gastrointestinal

[1]Department of Biochemistry, School of Medicine, Autonomous University of Madrid (UAM), Madrid, Spain   [2]Instituto de Investigaciones Biomédicas Alberto Sols, Consejo Superior de Investigaciones Científicas (CSIC)-UAM, Madrid, Spain   [3]Instituto de Investigación del Hospital Universitario de La Paz (IdiPAZ), Madrid, Spain   [4]Division of Biological Science, Graduate School of Science and Technology, Nara Institute of Science and Technology, Nara, Japan   [5]Department of Biological Chemistry and Pharmacology, The Ohio State University, Columbus, OH, USA   [6]Neuroscience Research Institute, The Ohio State University, Columbus, OH, USA

Correspondence: francesc.garcia@uam.es

tract and testes, and is a promising drug target for diabetes and cancer (10, 11, 12).

In contrast, serotonin receptor 6 (HTR6) is only expressed in brain, mostly in regions affecting cognition (13). Drugs targeting HTR6 hold promise for treatment of disorders such as anxiety, depression, eating disorders, schizophrenia, and Alzheimer's disease, and as memory enhancers (13). Serotonin binding to HTR6 stimulates $G_s$-adenylyl cyclase-cAMP signaling, and this is enhanced by Hh pathway activation (14). HTR6 also activates cAMP synthesis constitutively in a manner dependent on its ciliary targeting, and on neurofibromin (14, 15). Moreover, HTR6 activates other effectors, such as the CDK5 and mammalian target of rapamycin kinases (13, 15, 16, 17, 18). HTR6 localizes to neuronal cilia and regulates their length, morphology and composition, effects through which it affects cognition in a mouse Alzheimer's disease model (13, 19, 20, 21). Thus, HTR6 ciliary targeting is important for its functions.

The mechanisms underlying ciliary GPCR targeting are poorly understood. Cis-acting ciliary targeting sequences (CTSs) have been identified in some cases. For some GPCRs, these CTSs are located in their C-terminal tails (CTs). This is the case, among others, of Rhodopsin, SMO, and the D1 dopamine receptor (22, 23, 24). In other instances, however, the CTSs map to the third intracellular loops (IC3s). This was first found for SSTR3 and HTR6, and was later extended to others like MCHR1, GPR161, MC4R, or NPY2R (9, 25, 26, 27, 28).

For SSTR3 and HTR6, their CTSs were discovered by generating chimeras between these ciliary GPCRs and their non-ciliary relatives SSTR5 and HTR7. After extensive analyses, replacing the IC3s of SSTR5 or HTR7 with those of SSTR3 or HTR6, respectively, was found to suffice for ciliary targeting of the non-ciliary receptors. Furthermore, ciliary targeting of these chimeras was abolished by mutating to phenylalanine the first and last residues of Ax[AS]xQ motifs (hereafter referred to as A-Q motifs) present in the IC3s of both SSTR3 and HTR6 (25). In this study, another interesting observation was noted: the reverse pair of chimeras, in which the IC3s of SSTR3 and HTR6 were replaced by those of SSTR5 and HTR7, still accumulated in cilia, indicating that the newly discovered CTSs, albeit sufficient for ciliary targeting of non-ciliary receptors, were dispensable for ciliary targeting of SSTR3 and HTR6 themselves. This suggested, it was also noted, the presence of additional CTSs in these receptors (25). A subsequent study confirmed that HTR6 IC3 is dispensable for its ciliary targeting (21).

Herein, we report the identification and characterization of those missing CTSs. We show that, for both SSTR3 and HTR6, ciliary targeting occurs as long as either IC3, CT, or both, are present. Conversely, removal of both completely prevents their ciliary accumulation. We then identify the residues required for the function of these CTSs. For HTR6, an LPG motif is critical for C-terminal CTS (CTS2) function, whereas an RKQ motif is key for IC3 CTS (CTS1) function. Interestingly, we also find that the A-Q motif in HTR6-IC3 is not needed for CTS1 function, and elucidate why mutating A-Q to F-F indeed prevents ciliary targeting of the aforementioned chimera. In contrast, the tandem AP[AS]CQ motifs in SSTR3-IC3 do affect CTS1 function, in conjunction with a neighboring arginine-rich tract. On the other hand, SSTR3 CTS2 function mostly depends on the residues immediately following its seventh transmembrane helix, including LLxP and FK motifs, the latter homologous to the WR motif driving SMO ciliary targeting (29).

Finally, we studied how these newly identified CTSs control binding to well-established ciliary trafficking adapters such as TULP3 and RABL2B, both of which interact with the intraflagellar transport (IFT) machinery (30, 31, 32, 33). We show that HTR6 ciliary targeting is TULP3-dependent, as previously shown for SSTR3 and other ciliary GPCRs (27, 30, 34). Moreover, we find that the CTs of both HTR6 and SSTR3 associate with TULP3, in contrast to other TULP3-dependent GPCRs such as GPR161, whose association is IC3-dependent (34). For HTR6, we go on to show that TULP3 association is mediated by sequences near the LPG motif, which is itself not needed but rather antagonizes TULP3 association. Thus, TULP3 dissociation from HTR6 is likely to be an important step for the latter's ciliary accumulation. We further show that TULP3 is strongly needed for both CTS1 and CTS2 function in HTR6. Regarding RABL2B, which interacts with HTR6 and is required for its ciliary targeting (31), we find that, although it interacts with both IC3 and CT of HTR6, it is mostly required for CTS1 function, and only mildly affects CTS2 function.

# Results

## Ciliary targeting of HTR6 depends on TULP3

Many ciliary GPCRs depend on TULP3 for ciliary targeting. However, whether this is also true for HTR6 has not yet been determined. To clarify this issue, we used HTR6-IMCD3 cells, which we previously generated to stably express HTR6 (31). Expression of two independent TULP3 siRNAs (siTULP3 #1 and #2) in these cells caused a strong reduction in ciliary HTR6 intensity, which was not observed when a negative control luciferase siRNA (siLUC) was expressed instead (Fig 1A and B). Quantitative RT-PCR and Western blot confirmed that both siTULP3 #1 and #2 significantly reduced TULP3 mRNA and protein levels relative to siLUC (Fig 1C and D), and that HTR6 protein levels are unaffected by TULP3 knockdown, indicating that loss of HTR6 ciliary staining represents a ciliary targeting defect, rather than altered HTR6 protein synthesis or stability (Fig 1D). Thus, HTR6 joins the growing list of ciliary GPCRs known to depend on TULP3 for ciliary accumulation (34).

## HTR6-IC3 is sufficient but not necessary for HTR6 ciliary targeting

HTR6-IC3 contains a CTS, whose identification involved the study of chimeric GPCRs combining sequences of both HTR6 and HTR7, a non-ciliary HTR6 homolog (25). Before embarking on similar chimera studies, we first confirmed that HTR7 is indeed not a ciliary GPCR. Similar to the previous study, we found that HTR7-EGFP only localizes to ≈10% of cilia, as opposed to virtually 100% for HTR6-EGFP (Fig 2A–C) (25). Moreover, HTR7 levels in the few HTR7-positive cilia were much lower than HTR6 levels in HTR6-positive cilia (Fig S1). Thus, our data fully validate HTR7 as a good negative control for chimera studies of HTR6 ciliary targeting. Likewise, we fully confirmed previous results indicating that: (i) a chimera containing the first half of HTR6-IC3 (aa 208–241) in an HTR7 background (referred to here as Chimera N), accumulates in cilia much more efficiently than HTR7 (Fig 2A–C) (25); and (ii) cilia localization of Chimera N is disrupted by two point mutations (A230F+Q234F) in the ATAGQ motif of HTR6-IC3 (Fig 2A–C) (25). However, we also noticed that the A230F+Q234F

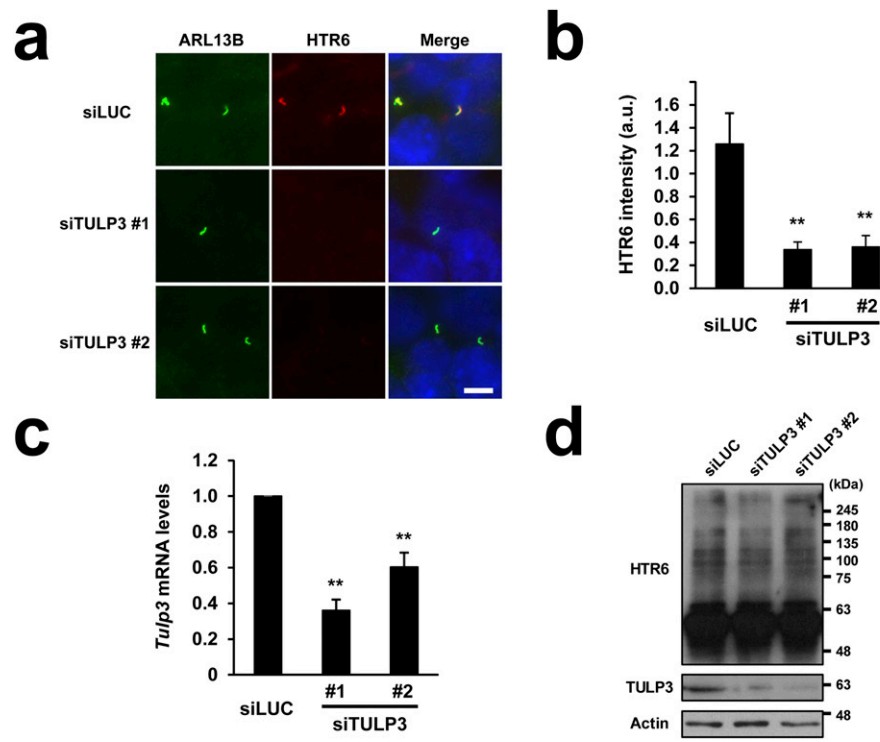

**Figure 1. HTR6 ciliary targeting requires the ciliary trafficking adapter TULP3.**
**(A)** HTR6-IMCD3 cells were transiently transfected with siRNAs targeting mouse TULP3 (siTULP3 #1 or siTULP3 #2) or firefly luciferase (siLUC) as negative control, serum-starved to promote ciliogenesis and immunostained with anti-ARL13B (green) and anti-HTR6 (red) antibodies. DNA was stained with Hoechst (blue). Scale bar, 5 $\mu$m. **(A, B)** HTR6 ciliary intensity was quantified from (A). Data are mean ± SEM of n = 23,32,29 cells for siLUC, siTULP3 #1, and siTULP3 #2, respectively. **(C)** Mouse Tulp3 mRNA levels were analyzed by qRT-PCR and expressed relative to Gapdh mRNA. Data are mean ± SEM of n = 3 independent experiments. Significance in (B, C) is shown as $P <$ 0.01(**) in unpaired two-tailed $t$ tests. **(A, D)** HTR6-IMCD3 cells transfected as in (A) were analyzed by Western blot with antibodies against HTR6, TULP3 and Actin, as indicated. Despite the siRNA-induced reduction in TULP3 protein, HTR6 protein levels remained unaltered. Molecular weight markers shown on the right.
Source data are available for this figure.

mutation increases Chimera N's intracellular retention by ≈7-fold, likely explaining this mutant's failure to accumulate in cilia (Fig 2D and E).

To determine whether A230F+Q234F also disrupts cilia localization of wild-type HTR6, we generated the HTR6(A230F+Q234F) mutant. Interestingly, this protein accumulates in cilia as efficiently as wild-type HTR6 (Fig 2A–C). This suggests that the ATAGQ motif, despite being necessary for ciliary targeting of Chimera N, is dispensable for ciliary targeting of HTR6. This is also in concordance with Berbari et al., who pointed out that HTR6 continues to accumulate in cilia when its HTR6-IC3 is replaced by HTR7-IC3 (25). Again, we fully confirmed ciliary targeting of this latter chimera (Chimera J) (Fig 3). Thus, HTR6 ciliary accumulation does not require HTR6-IC3.

### Ciliary targeting of HTR6 involves cooperation between IC3 and CT

Because Chimera J still accumulates in cilia, we reasoned that HTR6 must contain additional CTSs besides those in its IC3. Because C-terminal tails (CTs) of GPCRs are another common site for CTSs (22, 23), we tested whether HTR6-CT was necessary for ciliary accumulation of Chimera J. To do this, we generated Chimera O, which is identical to Chimera J except that it contains HTR7-CT instead of HTR6-CT (Fig 3A). When expressed in IMCD3 cells, Chimera O completely failed to accumulate in cilia, as opposed to Chimera J and wild-type HTR6 (Fig 3A–C). Lack of ciliary localization of Chimera O was not due to protein instability or retention in Golgi or ER, as Chimera O was readily seen at the plasma membrane (Fig S2). These data indicate that HTR6-CT functions as a CTS in the context of Chimera J.

To assess the relative contributions of HTR6-CT and HTR6-IC3 to ciliary targeting of HTR6, we created Chimera D, which contains HTR6-IC3 but lacks HTR6-CT (Fig 3A). Like Chimera N (Fig 2A–C), Chimera D also localized

to IMCD3 cilia, confirming that HTR6-CT, like HTR6-IC3, is dispensable for HTR6 ciliary targeting (Fig 3A and B). Altogether, these data indicate that HTR6 ciliary targeting involves redundancy between IC3 and CT: each is sufficient for HTR6 to accumulate in cilia, but none is individually required.

Although chimeras D and J are readily seen in cilia, their ciliary targeting is not as robust as that of wild-type HTR6. Upon quantitation, HTR6 robustly localized to virtually all cilia, chimeras D and J were present in about half of them, and Chimera O was completely absent from them (Fig 3C). These data seem to indicate that HTR6-IC3 and HTR6-CT are only partially redundant, as each alone is not sufficient for HTR6 cilia localization to be fully penetrant. However, as shown below, this is likely due to chimera use, as HTR6-CT inactivation by more specific point mutations has no such effects.

### Ciliary targeting function of HTR6-CT maps between residues 392–424

We next set about identifying which amino acid residues constitute HTR6's CTS2, that is, the CTS within HTR6-CT (Fig 3A). HTR6-CT (aa 326–440) is twice as long as HTR7-CT (Fig 4A). Because HTR6-CT's first half can be aligned with HTR7-CT, we generated another chimera by swapping the first half of HTR6-CT in Chimera J by HTR7-CT. The resulting protein, Chimera Q, still localized to cilia, indicating that aa 326–368 of HTR6-CT are not essential for CTS2 function (Fig 4A–D). Consistent with this, ciliary targeting of HTR6 was not affected by two mutations, S352A and S352D, which substitute non-phosphorylatable (Ala) and phosphomimetic (Asp) residues for Ser352, the mouse residue whose counterpart in human HTR6 is phosphorylated by CDK5 kinase (Fig S3) (17). Thus, the critical CTS2 residues should localize within HTR6-CT's second half. Accordingly, deletion of HTR6-CT's second half (Δ373-440) or last third (Δ401-440)

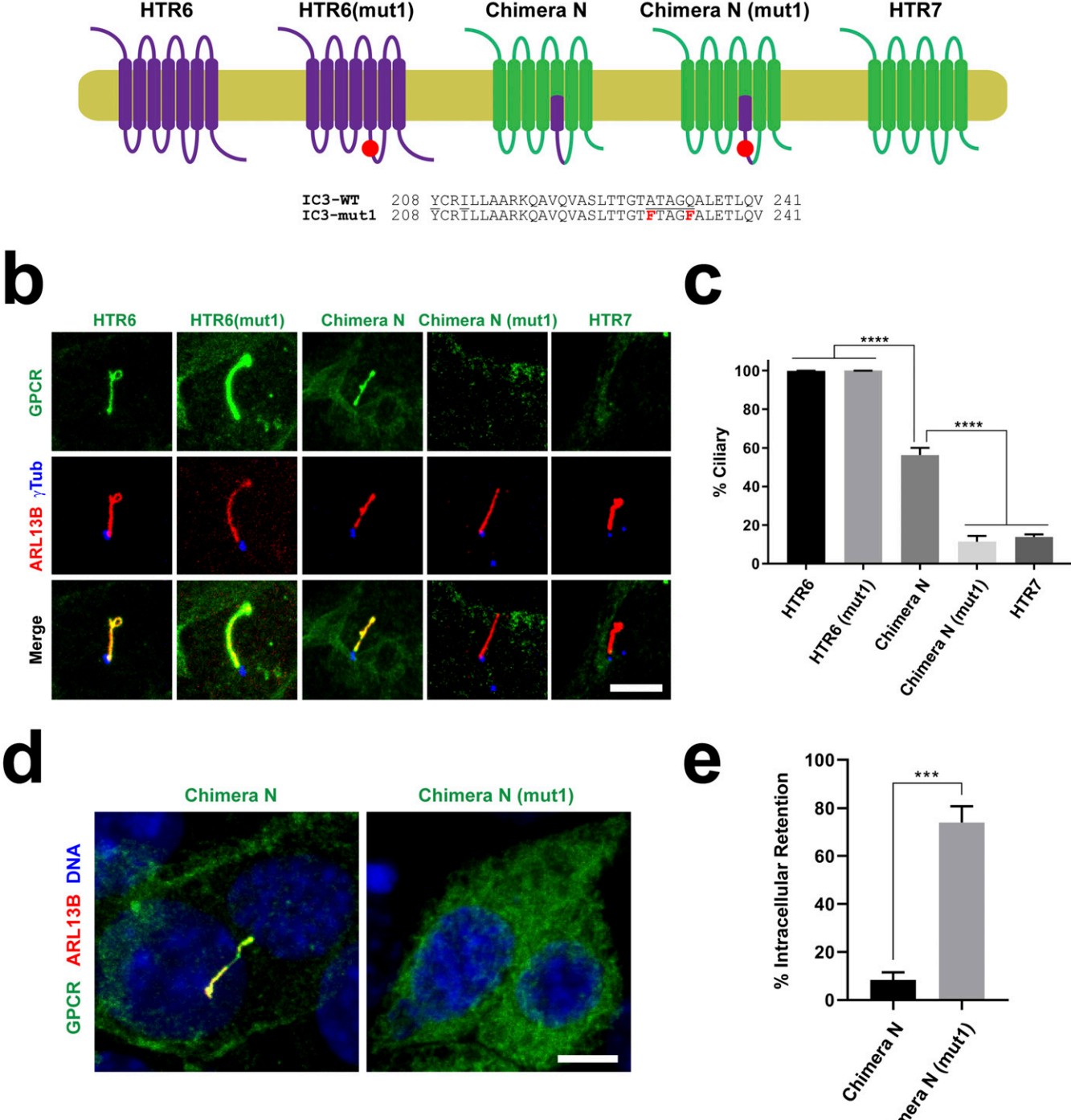

**Figure 2.   The ATAGQ motif in HTR6-IC3 is dispensable for ciliary targeting of wild-type HTR6.**
**(A)** Schematic representation of HTR7 (green), HTR6 (purple), Chimera N, and the mutant versions of the latter two, carrying the A230F+Q234F double mutation (mut1) in the first half of HTR6's IC3 loop, whose sequence is shown below. **(A, B)** The G protein–coupled receptors (GPCRs) from (A), with EGFP fused to their C-termini, were expressed in IMCD3 cells and their cilia localization was analyzed by immunofluorescence with antibodies against EGFP (green), ARL13B (red) and gamma-tubulin (γTub, blue). Scale bar, 5 $\mu$m. **(B, C)** Percentage of GPCR-positive cilia in GPCR-transfected cells was quantitated from (B). Data are mean ± SEM of n = 3 to 5 independent experiments per construct, in each of which at least 50 transfected-cell cilia were counted for each GPCR. Data were analyzed by one-way ANOVA followed by Tukey's multiple comparisons tests. Significance is indicated as $P < 0.0001$ (****). **(D)** Immunofluorescence pictures of Chimera N and Chimera N (mut1) showing the latter's intracellular retention. Scale bar, 5 $\mu$m. **(E)** Percentage of transfected cells where indicated chimera was retained intracellularly with no observable plasma membrane staining was quantitated from immunofluorescence experiments. Data are mean ± SEM of n = 3 independent experiments, each with at least 150 transfected cells counted per chimera. Significance in unpaired two-tailed $t$ test shown as $P < 0.001$ (***).

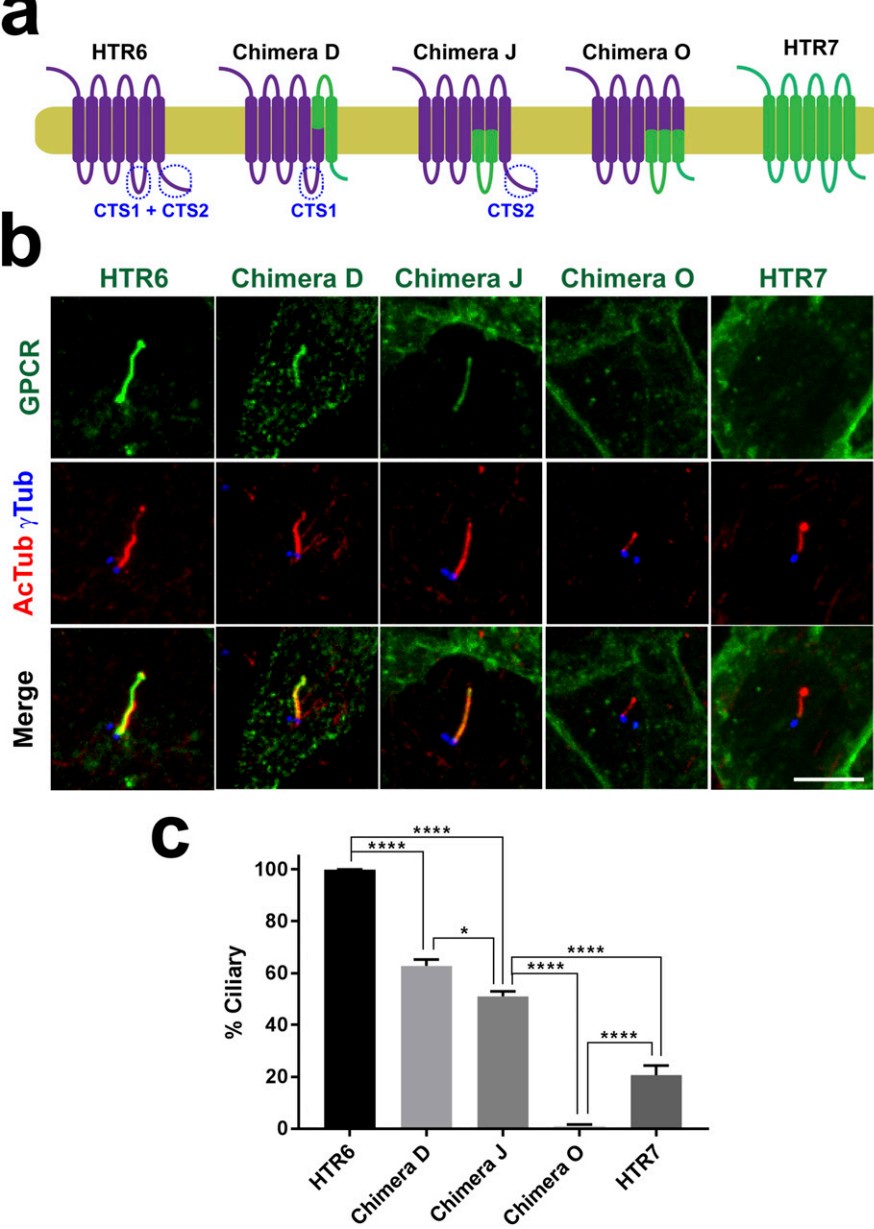

**Figure 3. HTR6 ciliary targeting relies on redundancy between IC3 loop (CTS1) and C-terminal tail (CTS2) sequences.**
**(A)** Schematic representation of HTR6 (purple), HTR7 (green) and their chimeras, wherein purple segments come from HTR6 and green ones from HTR7. Ciliary targeting sequences in IC3 loop (CTS1) and C-terminal tail (CTS2) are labeled where present. **(A, B)** The G protein-coupled receptors (GPCRs) from (A), with EGFP fused to their C-termini, were expressed in IMCD3 cells and their cilia localization was analyzed by immunofluorescence with antibodies against EGFP (green), acetylated tubulin (AcTub, red) and gamma-tubulin (γTub, blue). Scale bar, 5 μm. **(B, C)** Percentage of GPCR-positive cilia in GPCR-transfected cells was quantitated from (B) as described in the Materials and Methods section. Data are mean ± SEM of n = 5, 3, 5, 5, 8 (from left to right) independent experiments, in each of which at least 50 transfected-cell cilia were counted for each GPCR. One-way ANOVA followed by Tukey's multiple comparison tests shows all samples are significantly different from each other with $P < 0.0001$ (****) or $P < 0.05$ (*), as indicated.

completely abolishes cilia localization of Chimera J (Fig 4A–E), even though these mutants have no problem reaching the plasma membrane (Fig S2). In contrast, deletions Δ369-370, Δ371-378, Δ379-391, and Δ425-440 do not abolish ciliary targeting, even if Δ425-440, like Chimera Q, moderately reduces it (Fig 4A–E). Altogether, these data indicate that CTS2 function critically requires residues within aa 392–424, and that some residues outside this critical region reinforce CTS2 action.

### Ciliary targeting function of HTR6-CT is mediated by an LPG motif

To pinpoint which residues inside the critical region are required for CTS2 function, we generated nine alanine-scanning mutants (CT-mut1 to CT-mut9), together spanning all 33 residues in the region (Fig 4F). Seven of these mutants had no effect on cilia localization of Chimera J, whereas the other two mutants abolished it (Fig 4F–H). These two mutants, CT-mut3 (LLL398-400AAA) and CT-mut4 (PGE401-403AAA), still reached the plasma membrane, indicating that their absence from cilia is not due to lack of expression or failure to exit ER or Golgi (Fig S2). Next, we individually mutated to alanine each of the six residues covered by CT-mut3 and CT-mut4. Mutants L398A, L399A, and E403A did not significantly reduce ciliary targeting of Chimera J, whereas L400A, P401A, and G402A clearly did (Fig 4I and J). Thus, L400, P401, and G402 are the three key residues for HTR6 CTS2 function.

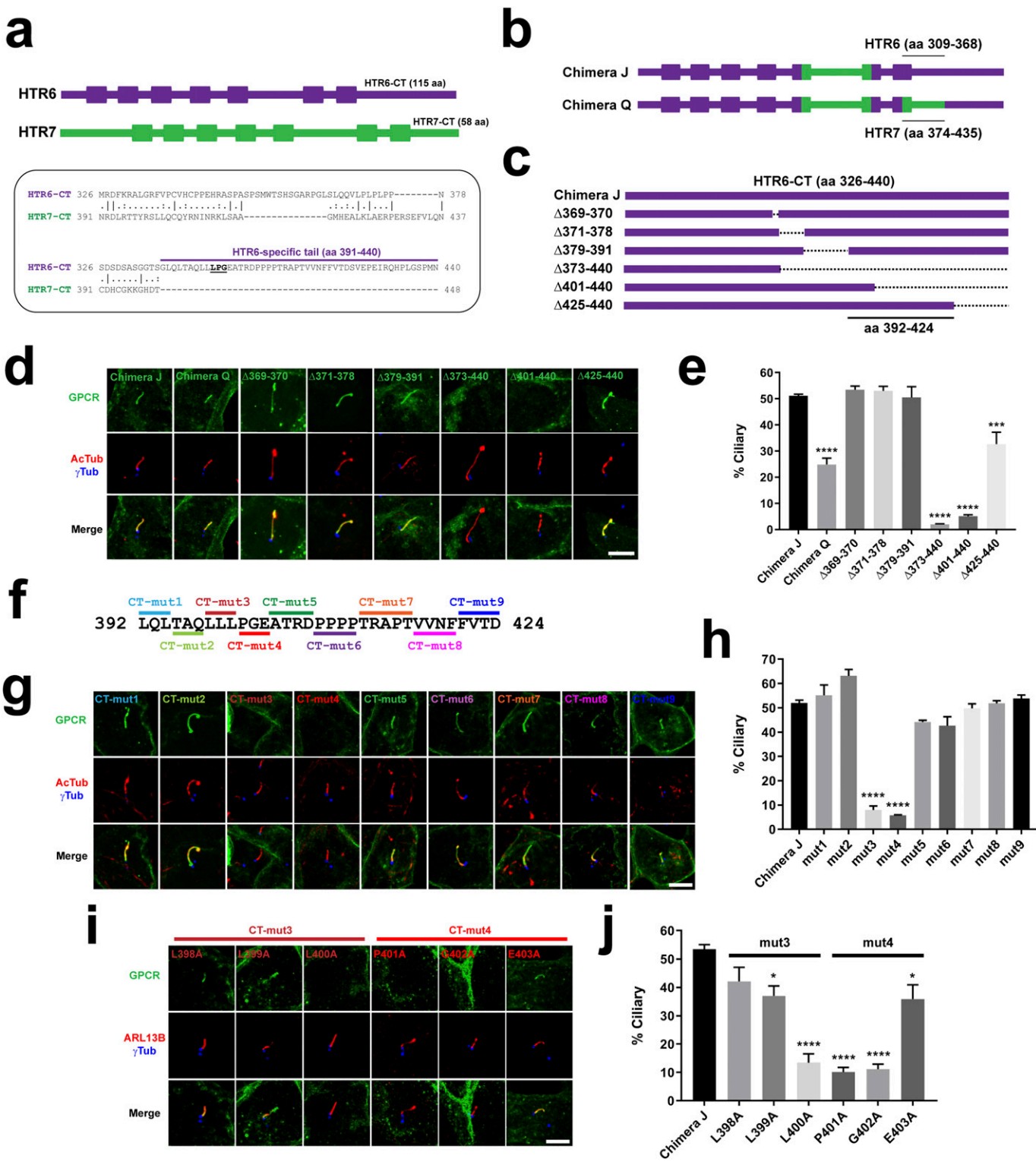

**Figure 4. An LPG motif is critical for the ciliary targeting function of HTR6's CTS2.**

**(A)** Top: schematic depiction of HTR6 (440 aa, purple) and HTR7 (448 aa, green) with transmembrane helices displayed as boxes. Notice how HTR6 C-terminal tail (CT) is twice as long as HTR7-CT (115 aa versus 58 aa). Bottom: alignment of HTR6-CT and HTR7-CT. The former is 10-fold richer in prolines (17% versus 1.7%) and its latter half (aa 391–440) has no homologous counterpart in HTR7. LPG motif inside HTR6-specific tail is underlined. **(B)** Schematic representation of Chimera J and Chimera Q. They are identical except that Chimera Q lacks the HTR6 residues indicated in Chimera J, and contains instead the HTR7 residues indicated in Chimera Q. **(C)** Schematic showing HTR6-CT on top (present in Chimera J), and the deletions that were introduced into Chimera J, covering all residues that had not been substituted in Chimera Q. Indicated at the bottom is the critical region required for ciliary targeting of Chimera J. **(B, C, D)** IMCD3 cells expressing the constructs from (B, C), all fused to EGFP in their C termini,

## Ciliary targeting function of HTR6-IC3 is mediated by an RKQ motif

Aside from the A230F+Q234F mutation in HTR6-IC3 interfering with Chimera N ciliary targeting (25) (Fig 2A–C), nothing is known about the exact residues mediating CTS1 function in HTR6-IC3. To clarify this, we first introduced the CT-mut3 mutation into wild-type HTR6 and checked its cilia localization, which was indistinguishable from wild-type HTR6 (Fig 5). Because CT-mut3 abolishes CTS2 function (Fig 4G and H), ciliary targeting of HTR6(CT-mut3) must depend on CTS1 function. Thus, we combined CT-mut3 with IC3 mutations to map CTS1 function (Fig 5A and B). Because HTR6-IC3 residues 208–241 are sufficient for ciliary targeting of Chimera N (Fig 2A–C) (25), we started by making three deletions spanning this sequence (Δ208-219, Δ220-229, and Δ230-241) and combining them with CT-mut3 (Fig 5A–D). The first two deletions (IC3-Δ1 and IC3-Δ2) abolished ciliary targeting of HTR6(CT-mut3), whereas the last deletion (IC3-Δ3) had no effect (Fig 5C and D). The strongest effect was seen for HTR6(IC3-Δ1+CT-mut3). Although this protein reaches the plasma membrane, it does so less efficiently, consistent with Δ208-219 disrupting a tyrosine-based sorting motif (208-YxxI-211) right after HTR6's fifth transmembrane helix (Figs 5A and S2) (35). For HTR6(IC3-Δ2+CT-mut3), the loss of ciliary targeting was strong but not complete, and this protein readily reached the plasma membrane (Figs 5C and D and S2). Thus, residues 208–229, but not 230–241, are important for CTS1 function.

The lack of effect by IC3-Δ3 in the CT-mut3 background was surprising, as IC3-Δ3 deletes the A-Q motif (230-ATAGQ-234), whose A230F+Q234F mutation (IC3-mut1) disrupts Chimera N ciliary targeting (Fig 2A–C) (25). Together with our observation that Chimera N(A230F+Q234F) is retained intracellularly (Fig 2D and E), this suggests that mistargeting of the latter is not due to absence of A230 and Q234, but rather to introduction of the two bulky and hydrophobic phenylalanines. Such dominant negative effects, however, were not seen with HTR6(IC3-mut1+CT-mut3), whose cilia localization is as good as that of wild-type HTR6 (Fig 5E and F). Thus, neither A-Q removal nor F-F introduction interfere with ciliary targeting of HTR6, even though F-F disrupts intracellular trafficking of Chimera N (Figs 2D and E and 5A–F).

To refine CTS1 mapping, we next created six alanine-scanning mutations (IC3-mut2 to IC3-mut7) spanning residues 208–229, with the exception of Y208 and I211 to avoid disrupting the aforementioned sorting motif (Fig 5A). Of these six mutants, four localized to cilia and two failed to do so (Fig 5E and F). These two mutants, IC3-mut4 (RKQ216-218AAA) and IC3-mut5 (VQV220-222AAA), were abundantly seen at the plasma membrane, showing that their effect on ciliary targeting is specific (Fig S2). Quantitatively, IC3-mut4 had a much stronger effect than IC3-mut5 (10-fold versus 1.7-fold reduction), indicating that 216-RKQ-218 are key for CTS1 function. As carried out above for CTS2, we individually mutated each residue in IC3-mut4

and IC3-mut5 to alanine. We also created the A219F mutant, so that the entire 216-RKQAVQV-222 stretch was covered. Analysis of IMCD3 cilia localization of these seven mutants showed that R216A and V222A reduce in half the ciliary targeting of HTR6(CT-mut3), whereas the other mutants do not affect it (Fig 5G and H). Thus, V222 fully accounts for the effect seen with IC3-mut5, whereas R216 only partially accounts for the stronger reduction seen with IC3-mut4. This suggests that K217 and/or Q218 positively contribute to CTS1 activity when R216 is absent. To test for this, we created the R216A+K217A and R216A+Q218A mutants, both of which phenocopy IC3-mut4 (Fig 5G and H). Thus, HTR6 CTS1 function critically depends on the RKQ triad, within which R216 is the most important residue, whereas K217 and Q218 play ancillary roles. Intriguingly, the RKQ triad is preceded by two alanines, making it a non-canonical A-Q motif (canonical being Ax[AS]xQ (25)). However, whether alanines 214–215 play a role in HTR6 CTS1 function remains to be explored.

## HTR6 CT and IC3 are both sufficient for ciliary targeting

Thus far, our data indicate that HTR6 cilia localization is mediated by cooperation between two redundant CTSs, CTS1 and CTS2, located in IC3 and CT, respectively. Therefore, in the context of HTR6-HTR7 chimeras or of HTR6 mutants, both CTS1 and CTS2 are sufficient to drive ciliary targeting. For CTS2, we further confirmed this by fusing HTR6-CT at the C-terminal end of HTR7 (Fig 6A). The resulting HTR7-(HTR6-CT) fusion protein strongly accumulates in cilia, showing that HTR6-CT is sufficient to target HTR7 to cilia (Fig 6B and C). We then tested whether HTR6-CT also suffices to target a single-pass transmembrane protein to cilia. To do this, we substituted HTR6-CT for the cytosolic domain of CD8α, a single transmembrane protein that has repeatedly been used for this same purpose (34, 36). Indeed, the CD8α-(HTR6-CT) chimera readily accumulated in cilia, as did CD8α-(HTR6-IC3). By contrast, CD8α-(HTR7-CT) failed to accumulate in cilia (Fig 6D–F), indicating that both HTR6-CT and HTR6-IC3 are sufficient to specifically target a single transmembrane protein to cilia. Last, we also checked whether HTR6-CT is sufficient to target a soluble protein to cilia. To test this, we created the (HTR6-CT)-EGFP fusion protein, which failed to accumulate in cilia, indicating that CTS2 function of HTR6-CT requires membrane association (Fig S4).

## SSTR3 ciliary targeting also involves redundant CTSs at IC3 and CT

In their study, Berbari et al showed that HTR6 ciliary targeting mechanisms resemble those of SSTR3 (25). Furthermore, SSTR3-IC3 was also seen to be sufficient but not necessary for cilia localization, as replacement of SSTR3-IC3 by the IC3 of non-ciliary SSTR5 did not abolish ciliary targeting (25). This observation, puzzling at

---

were analyzed by immunofluorescence with antibodies against EGFP (green), acetylated tubulin (AcTub, red) and gamma-tubulin (γTub, blue). Scale bar, 5 μm. **(D, E)** Percentage of G protein–coupled receptor (GPCR)–positive cilia relative to total transfected-cell cilia was quantitated from (D). **(C, F)** Nine separate mutations (CT-mut1 to CT-mut9) were introduced into critical region from (C), with all indicated residues replaced by alanines. **(D, G)** IMCD3 cells expressing C-terminally EGFP-tagged mut1-mut9 Chimera J mutants were analyzed as in (D). Scale bar, 5 μm. **(E, G, H)** Percentage of GPCR-positive cilia from (G) was quantitated as in (E). **(D, I)** The residues mutated in CT-mut3 and CT-mut4 were individually substituted to alanine and analyzed as in (D). **(E, I, J)** Percentage of GPCR-positive cilia from (I) was quantitated as in (E). In all quantitations, data are mean ± SEM of n = 3–5 independent experiments per construct, with at least 50 cilia counted per construct and experiment. Statistical analysis was performed by one-way ANOVA followed by Tukey's multiple comparisons tests. Significance is indicated as $P < 0.05(*)$, $P < 0.001(***)$, or $P < 0.0001(****)$.

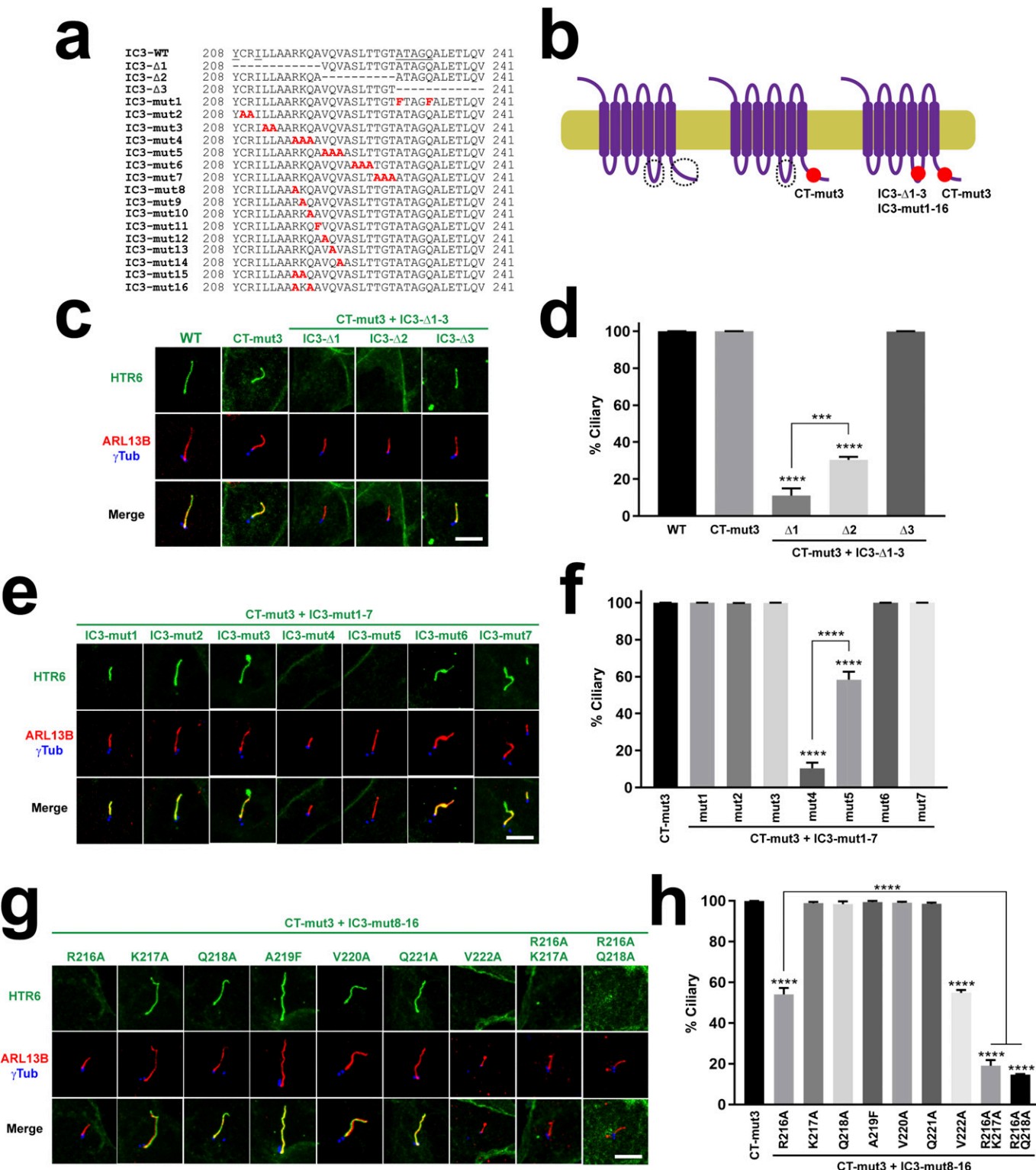

**Figure 5. An RKQ motif is critical for IC3-dependent HTR6 ciliary targeting.**
**(A)** Sequence of HTR6's IC3 loop and its mutants used here. **(B)** Schematic of HTR6 wild type and its mutants used here. CT-mut3 is the mut3 CTS2 mutation from Fig 3. The IC3 mutations from (A) were combined with CT-mut3. Mutations shown as red spots. CTS1 and CTS2 encircled with dashed lines when intact. **(C)** IMCD3 cells expressing the indicated versions of HTR6, all fused to C-terminal EGFP, were analyzed by immunofluorescence with antibodies against EGFP (green), ARL13B (red) and gamma-tubulin (γTub, blue). Scale bar, 5 $\mu$m. **(C, D)** Percentage of G protein-coupled receptor-positive cilia relative to total transfected-cell cilia was quantitated for the constructs in (C). **(C, E)** The indicated HTR6 mutants were analyzed as in (C). **(D, E, F)** Ciliary targeting of HTR6 mutants from (E) was quantitated as in (D). **(C, G)** The indicated HTR6 mutants were analyzed as in (C). **(D, G, H)** Ciliary targeting of HTR6 mutants from (G) was quantitated as in (D). Data in (D, F, H) are mean ± SEM of n = 3–4 independent experiments per construct, with at least 50 cilia counted per construct and experiment. Data were analyzed by one-way ANOVA followed by Tukey's multiple comparisons tests. Unless otherwise indicated, significance is shown relative to control sample (black column) with $P < 0.001$(***) or $P < 0.0001$(****).

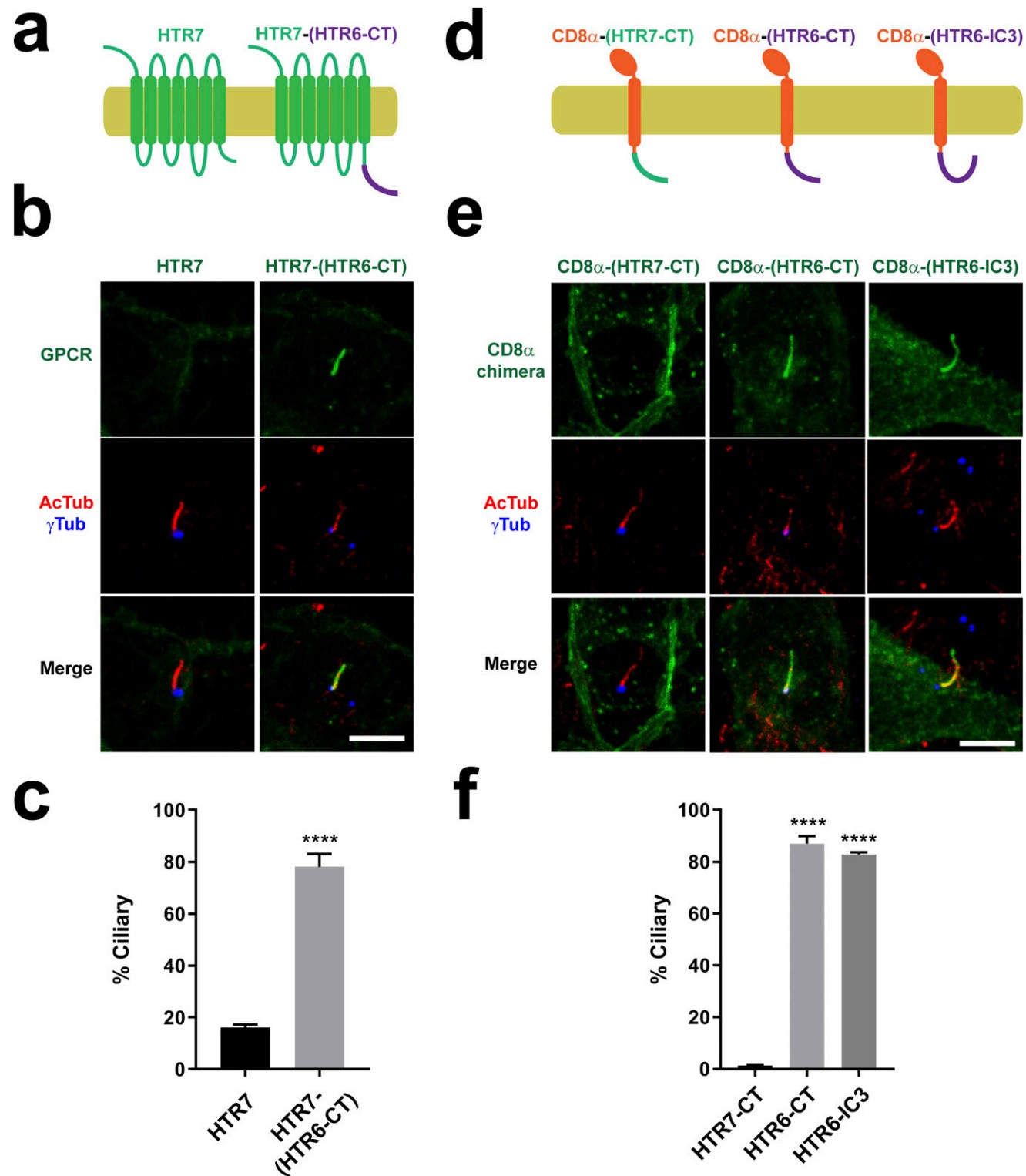

**Figure 6. HTR6 CTS1 and CTS2 are both sufficient for ciliary targeting.**
**(A)** Schematic of HTR7 with or without the CTS2-containing C-terminal tail of HTR6 (HTR6-CT) fused to its C terminus. **(A, B)** IMCD3 cells expressing C-terminally EGFP-tagged constructs from (A) were analyzed by immunofluorescence with antibodies against EGFP (green), acetylated tubulin (AcTub, red) and gamma-tubulin (γTub, blue). Scale bar, 5 μm. **(B, C)** Percentage of G protein-coupled receptor-positive cilia relative to total transfected-cell cilia was quantitated from (B). **(D)** Schematic of CD8α(1-206) chimeras, containing extracellular and transmembrane regions of CD8α fused to HTR7-CT, HTR6-CT (containing CTS2) or HTR6-IC3 (containing CTS1). **(B, D, E)** IMCD3 cells expressing C-terminally EYFP-tagged constructs from (D) were analyzed as in (B). Scale bar, 5 μm. **(E, F)** Percentage of G protein–coupled receptor-positive cilia relative to total transfected-cell cilia was quantitated from (E). Data in (C, F) are mean ± SEM of n = 4–5 independent experiments per construct. In each experiment, at least 50 cilia were counted per condition. **(C, F)** Data were analyzed by unpaired two-tailed $t$ test (C) or by one-way ANOVA followed by Tukey's multiple comparison tests (F). Significance in both cases is shown as $P < 0.0001$ (****).

the time, may now readily be explained if SSTR3-CT also contains a CTS2. To test this, we first checked whether SSTR3-CT is sufficient to drive cilia localization of a CD8α-(SSTR3-CT) chimera. Indeed, both CD8α-(SSTR3-CT) and CD8α-(SSTR3-IC3) chimeras accumulated in cilia, with the former doing so even more strongly than the latter (Fig 7A–C). Therefore, both SSTR3-IC3 and SSTR3-CT contain CTSs.

## Ciliary targeting function of SSTR3-CT is mediated by its juxtamembrane region

As performed with HTR6, we then characterized how the two CTSs functionally relate to one another and which residues underlie their function. For this, we generated a series of deletion and point mutants in both SSTR3-IC3 and SSTR3-CT (Fig 7D–F). Previously, a mutant was already identified in SSTR3-IC3 that disrupts ciliary targeting of an SSTR3-SSTR5 chimera containing SSTR3-IC3 in an SSTR5 background (25). When this mutation (A243F+Q247F+A251F+Q255F, henceforth IC3-mut1, Fig 7D) was introduced to wild-type SSTR3, cilia localization was only very mildly affected, reducing the percentage of SSTR3-positive cilia from ≈100 to ≈80% (Fig 7G and H). We then examined whether ciliary targeting of SSTR3(IC3-mut1) is disrupted by mutations in SSTR3-CT (aa 326–428) (Fig 7E and F). Deleting the last third of SSTR3-CT (CT-Δ1: Δ389-428) from SSTR3(IC3-mut1) caused another mild reduction in ciliary targeting, from ≈80 to ≈65% (Fig 7G and H). Deleting the central third of SSTR3-CT (CT-Δ2: Δ355-388), which contains a coiled coil highly enriched in glutamate residues, had a stronger effect, reducing targeting from ≈80 to ≈50% (Fig 7G and H). A very similar effect was observed with CT-Δ3 (Δ349-428), which gets rid of all but the first 23 aa of SSTR3-CT (Fig 7G and H). Thus, although SSTR3-CT aa 349–428 modulate ciliary targeting, they are not critical for it.

A mutation deleting aa 335–428 prevented ciliary but also plasma membrane targeting and was accumulated intracellularly (Fig S5). Instead, internal deletion of aa 335–348 (CT-Δ4) completely abolished ciliary targeting without affecting trafficking to plasma membrane (Figs 7F and G and S5). Deleting either the first (CT-Δ5: Δ335-RILLRP-340) or second (CT-Δ6: Δ341-SRRIRSQE-348) half of this sequence also abolished ciliary targeting, leaving only ≈20% of mildly positive cilia, the same as in the non-ciliary SSTR5 negative control (Fig 7G and H) (25). None of these mutations obstructed plasma membrane trafficking (Fig S5). Hence, the region 335–348 contains essential residues for CTS2 function of SSTR3. Among these residues there is an LLxP motif reminiscent of Rhodopsin's VxP and other CTSs (5, 24). Mutation of these three residues to alanine (CT-mut1: L337A+L338A+P340A) also abolished CTS2 function without causing other trafficking defects (Figs 7G and H and S5). In the immediate juxtamembrane region (aa 326–334), we also introduced the F329A+K330A mutation (CT-mut2), as disruption of this aromatic–basic pair prevents ciliary targeting of SMO and other GPCRs (29). Indeed, these residues are also needed for SSTR3's CTS2 function but not for plasma membrane targeting (Figs 7F and G and S5).

## Ciliary targeting function of SSTR3-IC3 is mediated by AP[AS]CQ motifs and a basic stretch

After identifying critical residues for CTS2 function of SSTR3-CT, we focused on the CTS1 function of SSTR3-IC3 (aa 231–266). As

expected, the CT-mut1 mutation alone did not prevent ciliary targeting when introduced into wild-type SSTR3 (Fig 7G and I). We then combined CT-mut1 with seven mutations spanning the entire SSTR3-IC3 (Fig 7D). The first two, IC3-mut2 (VVK231-233AAA) and IC3-Δ1 (Δ234-242), had no effect on SSTR3 CTS1 function (Fig 7G and I). We then tested the effect of residues 243–255, containing both AP[AS]CQ motifs (243-APSCQWVQAPACQ-255). IC3-Δ2 (Δ243-247) and IC3-Δ3 (Δ248-255) also had no detectable effect, perhaps because of redundancy between the motifs (Fig 7G and I). To test this, we looked at IC3-Δ4 (Δ243-255). Intriguingly, the IC3-Δ4+CT-mut1 protein was still present in 80% of cilia, which, albeit significantly lower than 100% in CT-mut1, is much higher than 10% in IC3-mut1+CT-mut1, the quadruple phenylalanine mutant (Fig 7G–I). Because both IC3-mut1+CT-mut1 and IC3-Δ4+CT-mut1 readily reach the plasmalemma (Fig S5), this suggests the phenylalanines in IC3-mut1 have dominant negative effects specifically affecting SSTR3 ciliary targeting. That is not the whole story, however. Not only did IC3-Δ4 lower the percentage of positive cilia (Fig 7I), but also visibly reduced the amount of ciliary staining (Fig 7J). Quantitation of ciliary signal intensity revealed a 70% decrease of IC3-Δ4+CT-mut1 relative to CT-mut1 control (Fig 7J). Thus, the AP[AS]CQ motifs region does indeed play an important role in SSTR3 CTS1 function.

The second A-Q motif is immediately followed by an arginine-rich stretch (256-RRRRSERR-263), whose deletion (IC3-Δ5) causes a twofold reduction in both ciliary presence and intensity (Fig 7G–J). Nevertheless, we also noticed that around 40% of IMCD3 cells expressing this mutant fail to traffic it to the cell surface, which may partially or fully explain its reduced ciliary targeting (Figs 7K and S5). No such intracellular retention was observed with IC3-Δ4+CT-mut1 or CT-mut1 control, where virtually all cells display prominent plasma and/or ciliary membrane localization of these proteins. The last mutant, IC3-mut3, affecting the three residues adjacent to SSTR3's sixth transmembrane helix, only had a minor effect (Fig 7G–J). Altogether, these data indicate that CTS1 function in SSTR3-IC3 is encoded by the AP[AS]CQ motif region and the subsequent arginine-rich stretch, even though the latter acts, at least partly, by enabling transport to the cell surface.

## Ciliary targeting of HTR6 and SSTR3 in neurons also involves CTS1 and CTS2 redundancy

All our ciliary trafficking analyses thus far were performed in the kidney epithelial IMCD3 cell line. However, HTR6 and SSTR3 are above all neuronal GPCRs, with HTR6 being expressed exclusively, and SSTR3 mostly, in the CNS, where both receptors accumulate in neuronal cilia (5, 13). Thus, our IMCD3 data would be of little significance if they were not also applicable to ciliated neurons in the CNS. To test this, we expressed HTR6 and SSTR3, or mutants lacking one or both of our identified CTSs, in primary hippocampal neuron cultures (Fig 8). In these neurons, wild-type HTR6 was always clearly detectable in cilia (nine positive cilia out of nine ciliated and transfected cells: 9/9). For Chimera J, lacking CTS1, the proportion was 9/10. For HTR6-(CT-mut3), lacking CTS2, it was 10/10. In contrast, for Chimera J-(CT-mut3), lacking both CTS1 and

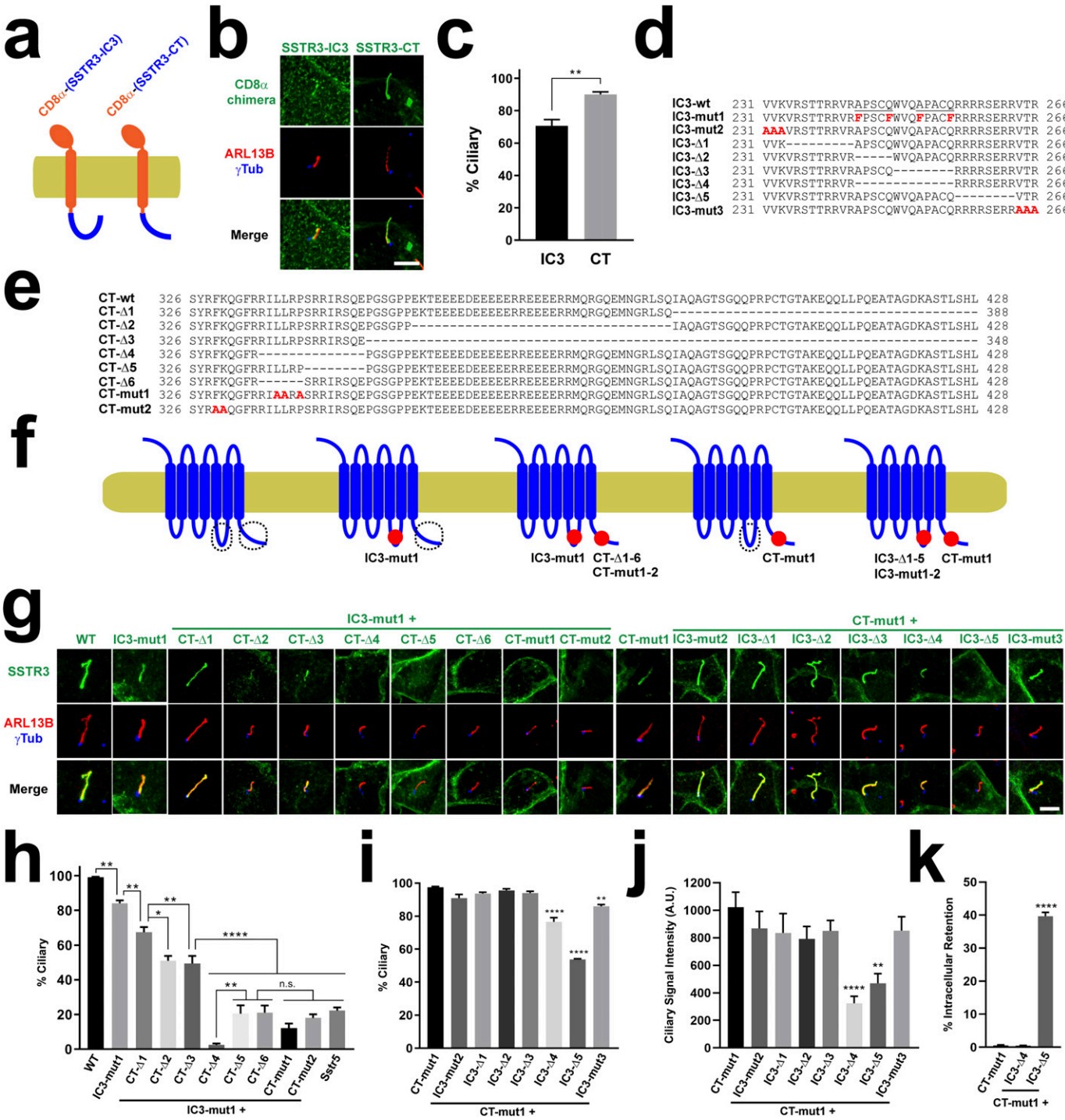

**Figure 7. SSTR3 ciliary targeting also depends on redundant ciliary targeting sequences in IC3 and CT.**
**(A)** Schematic of CD8α(1-206) chimeras fused to SSTR3-IC3 or SSTR3-CT. **(A, B)** IMCD3 cells expressing C-terminally EYFP-tagged constructs from (A) were analyzed by immunofluorescence with antibodies against EGFP/EYFP (green), ARL13B (red), and gamma-tubulin (γTub, blue). Scale bar, 5 μm. **(B, C)** Percentage of G protein-coupled receptor-positive cilia relative to total transfected-cell cilia was quantitated for the constructs in (B). Data are mean ± SEM of n = 4, 8 (IC3, CT) independent experiments per construct. Significance in unpaired two-tailed t test shown as P < 0.01(**). **(D)** SSTR3-IC3 wild-type sequence (top) and its mutated versions used below. The two reported Ax(A/S)xQ motifs are underlined in wild-type sequence. **(E)** SSTR3-CT wild-type sequence (top) and its mutated versions used below. **(F)** Schematic of SSTR3 and its mutants used below. CTS1 and CTS2 are encircled where intact. Mutations shown as red spots. **(B, F, G)** Ciliary targeting of SSTR3 mutants from (F) was analyzed as in (B). **(C, G, H, I)** Percentage of positive cilia for each of the indicated SSTR3 constructs from (G) was quantitated as in (C). **(J)** Intensity of ciliary staining was quantitated for the indicated SSTR3 constructs. **(K)** Percentage of cells with no detectable plasma membrane or ciliary staining was quantitated for indicated constructs. Data in (H, I, J, K) are mean ± SEM and were analyzed by one-way ANOVA followed by Tukey's multiple comparison tests. Significance shown as P < 0.05(*), P < 0.01(**), P < 0.001(***), P < 0.0001(****) or not significant (n.s.). For (H), numbers of independent experiments per construct from left to right were n = 10, 3, 4, 3, 3, 3, 4, 4, 3, 4, 10. Equivalent numbers for (I) were n = 3, 4, 4, 5, 5, 5, 4, 4. For both (H, I), at least 50 cilia were counted per construct and experiment. For (J), intensity was measured in n = 26–59 cilia per condition in one representative experiment. For (K), n = 4 independent experiments per construct with at least 200 transfected cells assessed per construct and experiment.

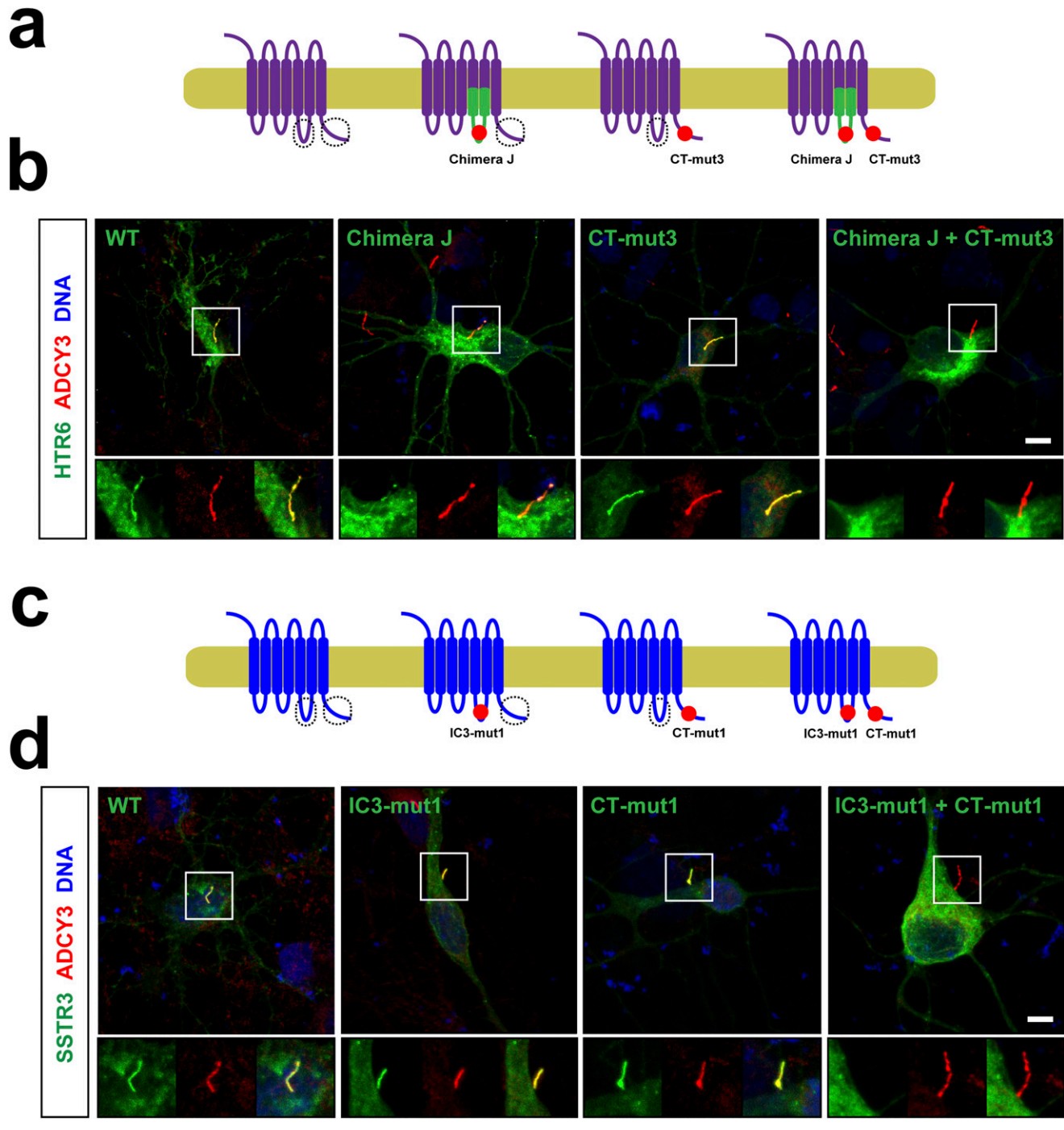

**Figure 8. Ciliary targeting of HTR6 and SSTR3 in hippocampal neurons also depends on redundancy between CTS1 and CTS2.**
**(A)** Schematic of the HTR6 constructs used here. **(A, B)** Cultured hippocampal neurons expressing C-terminally EGFP-tagged constructs from (A) were analyzed by immunofluorescence with antibodies against adenylate cyclase 3 (ADCY3, red). DNA was stained with DRAQ5 (blue) and EGFP fluorescence was directly visualized. **(C)** Schematic of the SSTR3 constructs used here. **(B, C, D)** Cultured hippocampal neurons from (C) were analyzed as in (B). Scale bars, 10 $\mu$m.

CTS2, the fraction was 2/16 (Fig 8A and B). For SSTR3, the proportions were 9/10 for wild-type, 8/10 for IC3-mut1 (lacking CTS1 function), 8/8 for CT-mut1 (lacking CTS2), and 2/12 for the double mutant IC3-mut1+CT-mut1 (Fig 8C and D). Therefore, ciliary targeting of HTR6 and SSTR3 in hippocampal neurons follows the same rules as in IMCD3 cells.

**HTR6 and SSTR3 CTs associate with TULP3**

Because ciliary accumulation of both HTR6 (Fig 1) and SSTR3 (30) depend on TULP3, which associates to the IC3 loops of two ciliary GPCRs, GPR161, and MCHR1 (34), we then examined whether HTR6 and SSTR3 also interact with TULP3 through their IC3s and/or CTs. As

controls, we used the IC3s and CTs of a non-ciliary GPCR (β2AR) and of the aforementioned GPR161 (34). Because TULP3 association to ciliary GPCRs has been detected by proximity biotinylation assays, we performed these experiments as previously described, except for the use of an improved biotin ligase, BioID2 (Fig 9A and B) (34, 37). As expected, GPR161-IC3 but not β2AR-IC3 robustly associates with TULP3 (Fig 9C and D). Interestingly, neither HTR6-IC3 nor SSTR3-IC3 associated with TULP3 (Fig 9C and D). In contrast, GPR161-CT showed no specific TULP3 association when compared to β2AR-CT, whereas both HTR6-CT and SSTR3-CT associated to TULP3 (Fig 9C and E). Whereas SSTR3-CT association was very strong (20-fold over β2AR-CT control, P < 0.0001), HTR6-CT association was less intense (fourfold increase, P = 0.069), but still faithfully reproduced in all five independent experiments we performed (Fig 9C and E). Thus, it appears that different ciliary GPCRs associate with TULP3 in different ways: some rely mostly on their IC3s, whereas others use their CTs.

We also determined which of TULP3's two domains is involved in its association with ciliary GPCRs. TULP3 N-terminal domain, which binds to the IFT-A ciliary trafficking complex (30), did not show any association to HTR6-CT (Fig S6). On the other hand, TULP3 C-terminal domain, its phosphoinositide-binding Tubby domain, readily associated with HTR6-CT (Fig S6), consistent with previous data showing that TULP3 association to ciliary GPCRs requires the former's ability to bind phosphoinositides (34).

### HTR6 CTS2 antagonizes TULP3 association

We then used proximity biotinylation to test whether CTS2 mutation in HTR6-CT affects TULP3 association. To our surprise, the CT-mut3 and Δ390-407 mutations, both lacking the LPG motif essential for CTS2 function (Fig 4), displayed much stronger TULP3 association than wild-type HTR6-CT (Fig 10). CT-mut3 also clearly increased HTR6-CT association to TULP3's C-terminal Tubby domain (Fig S6). In contrast, deleting aa 373–389, which are dispensable for cilia localization of Chimera J (Fig 4), completely abolished TULP3 association, as did the bigger Δ373-440 deletion (Fig 10). Thus, aa 373–389 promote TULP3 binding, whereas aa 390–407 antagonize it. Interestingly, these effects appear to cancel each other out in HTR6-CT(Δ373-407), whose TULP3 association resembles that of wild type (Fig 10). Finally, HTR6-CT(Δ408-440) also behaves like HTR6-CT(WT), indicating that aa 408–440 are not needed for TULP3 association with HTR6-CT, even though adding these residues to HTR6-CT(Δ373-440) rescues its TULP3 association in HTR6-CT(Δ373-407) (Fig 10). Altogether, these data suggest that strong TULP3 association is not essential for HTR6 ciliary targeting. Instead, preventing excessive TULP3 association, or promoting its dissociation, may be more important for HTR6 ciliary accumulation.

### TULP3 regulates HTR6 ciliary targeting through both HTR6-IC3 and HTR6-CT

We then tested whether TULP3 is necessary for CTS1 and CTS2 function. To do this, we first used CRISPR to generate TULP3-null IMCD3 clones. Because TULP3 is needed for ciliary targeting of ARL13B, we used lack of this ciliary marker as a way to identify likely TULP3 loss of function clones (38). In this way, we identified clones

#2-10 and #3-7. Genomic analysis of clone #2-10 revealed two different *Tulp3* alleles, both completely lacking exon 2, leading to frameshift after residue 12 (Fig 11A). In clone #3-7, we identified three alleles, two of which were clearly null, whereas the last one contained an in-frame deletion of residues 20–24, which removes part of the α-helix required for TULP3 binding to the IFT-A complex, an interaction that is required for ciliary GPCR targeting (Fig 11A) (30). Because ARL13B could not be used to look at cilia in these clones, we tried staining them for acetylated α-tubulin (AcTub), another widely used ciliary marker. However, for reasons that are not yet clear to us, AcTub staining did not work well in these clones. We therefore tried another ciliary marker, IFT88, which did label cilia in these clones (Fig 11B). Quantitation of ciliation frequencies showed no differences between wild-type and TULP3 mutant cells (Fig 11C).

We then used these clones to assess how TULP3 affects ciliary targeting of CD8α-(HTR6-IC3) and CD8α-(HTR6-CT) (both also containing Stag-TEV-EYFP in their C-termini) (Fig 11D–G). In wild-type cells, about 80% of transfected and ciliated cells contained medium-to-high levels of CD8α-(HTR6-IC3). In contrast, this construct was never seen at high levels in TULP3 mutant cilia, and even medium intensity was very rare, with more than 90% of transfected cell cilia having either low or undetectable CD8α-(HTR6-IC3) levels (Fig 11D and E). Virtually the same results were obtained when CD8α-(HTR6-CT) was transfected instead (Fig 11F and G). These highly significant effects demonstrate that TULP3 is required for both CTS1 and CTS2 function.

### RABL2 interacts with both HTR6-IC3 and HTR6-CT

RABL2 is an atypical RAB small GTPase that promotes anterograde IFT from the ciliary base (31, 32, 33, 39). Whereas the mouse genome contains a single *Rabl2* gene, the human genome contains two closely related paralogs, *RABL2A* and *RABL2B* (31, 32, 33). Recently, we showed that RABL2 interacts and is required for ciliary targeting of HTR6 and GPR161 (31). Because HTR6 and RABL2B interact by coimmunoprecipitation (co-IP), we tested whether their interaction depends on our newly identified CTSs. As expected, EGFP-RABL2B robustly co-immunoprecipitated (co-IPed) myc-HTR6 when both were expressed in HEK293T cells (Fig 12A). This was not noticeably affected when myc-HTR6(CT-mut3), lacking CTS2, was used instead. In contrast, co-IP of myc-Chimera J, lacking HTR6-IC3, and thus also CTS1, was strongly reduced, as was co-IP of myc-Chimera J (CT-mut3), which appeared even lower (Fig 12A). These data were confirmed in similar co-IP experiments using Flag-RABL2B instead of EGFP-RABL2B (Fig S7). This suggests that HTR6-IC3 is important for HTR6-RABL2B binding and that CTS2 may contribute to the interaction in the context of Chimera J, but not of wild-type HTR6.

We then checked the effects of mutating CTS1 within HTR6-IC3. To test this, we used the IC3-mut4 mutation removing the critical RKQ motif. EGFP-RABL2B co-IPed myc-HTR6(IC3-mut4), myc-HTR6(CT-mut3), and myc-HTR6(IC3-mut4+CT-mut3) to a similar extent than myc-HTR6(WT) (Fig 12B). Thus, neither CTS1 nor CTS2 play a major role in RABL2B binding to full length HTR6.

In contrast, IC3-mut4 strongly reduced co-IP of CD8α-(HTR6-IC3)-myc by EGFP-RABL2B (Fig 12C). EGFP-RABL2B also strongly co-IPed CD8α-(HTR6-CT)-myc, an interaction that was not impaired by CT-mut3 (Fig 12C). Of note, both IC3-mut4 and CT-mut3 reduced

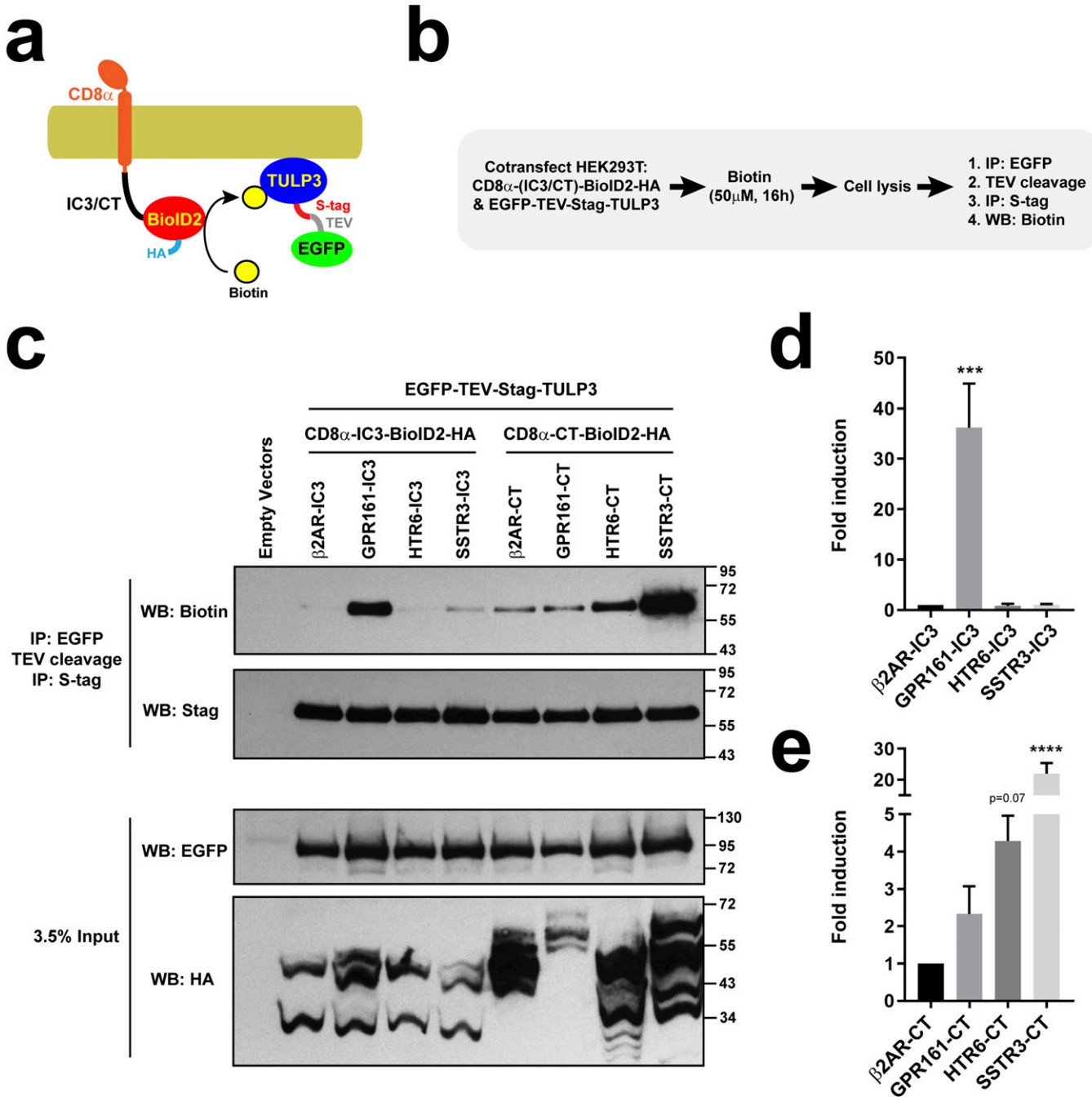

**Figure 9. HTR6 and SSTR3 CTs associate with ciliary trafficking adapter TULP3.**

**(A, B)** Schematic and protocol of BioID2 proximity labeling assay. HEK293T cells were cotransfected with plasmids encoding EGFP-TEV-Stag-TULP3 and a fusion protein containing the extracellular and transmembrane regions of CD8α (aa 1–206), the C-terminal tail (CT) or third intracellular loop (IC3) of a G protein-coupled receptor, the BioID2 biotin ligase, and an HA epitope. In presence of biotin (50 μM, 16 h), BioID2 biotinylates surrounding proteins in a proximity-dependent manner. After cell lysis, TULP3 was affinity purified by two sequential immunoprecipitations (IP) and its biotinylation assessed by Western blot (WB). **(C)** SDS–PAGE and WB analysis of immunoprecipitated S-tagged TULP3 (top two panels) and of the cleared cell lysates (bottom two). In the IPs, NeutrAvidin-HRP was used to detect TULP3 biotinylation (top) and anti-Stag antibody to detect its total levels. In the lysates, anti-EGFP and anti-HA tag antibodies were used to detect EGFP-TEV-Stag-TULP3 and the CD8 fusions, respectively. Molecular weight markers are indicated on the right (kD). **(C, D, E)** Biotinylated TULP3 signal, relative to total TULP3 signal in IPs, was quantitated from n = 5 independent experiments like the one in (C). Biotinylation by IC3 constructs (D) and by CT constructs (E) was normalized relative to β2AR-IC3 and β2AR-CT, respectively. Data are mean ± SEM and were analyzed by one-way ANOVA followed by Dunnett's multiple comparison tests. Significance shown as $P < 0.0001$ (****). Source data are available for this figure.

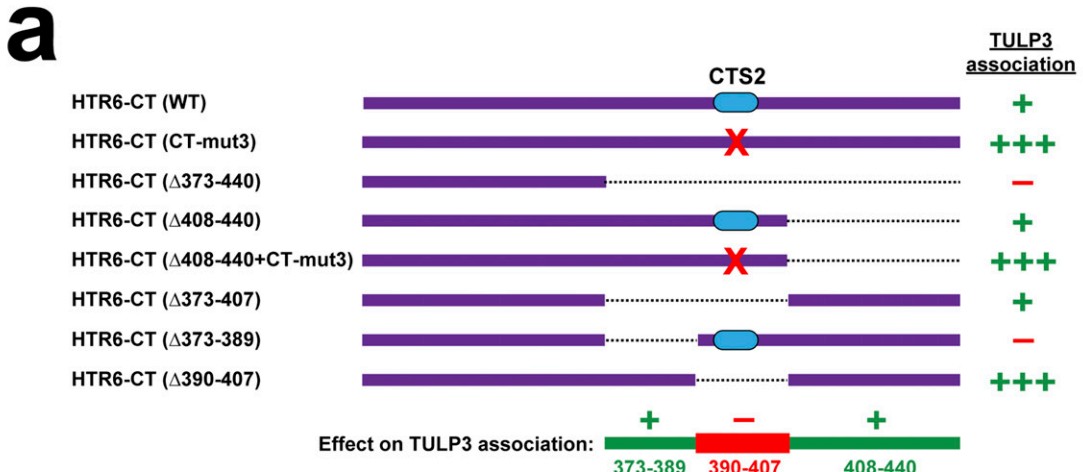

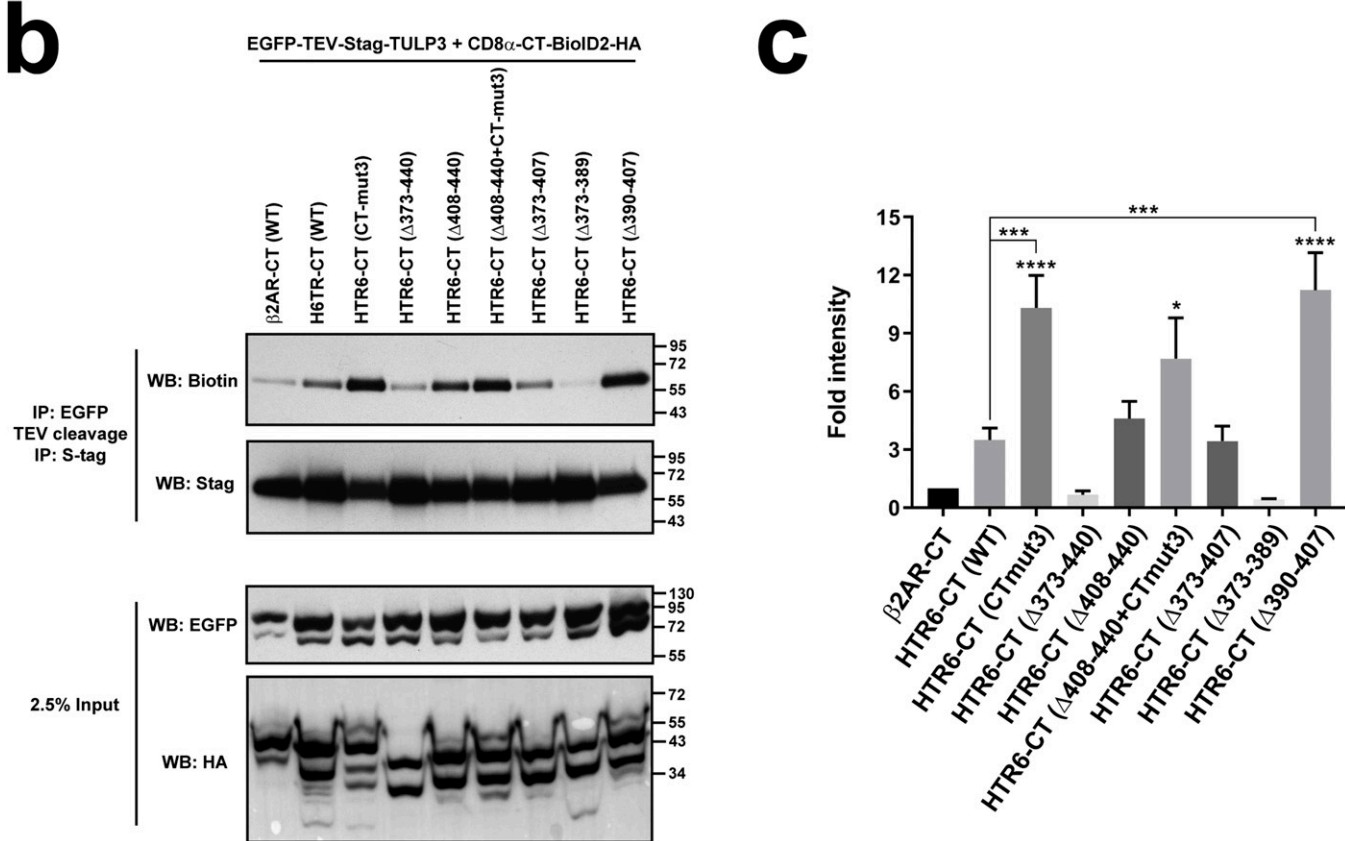

**Figure 10. HTR6 CTS2 antagonizes TULP3 association to HTR6-CT.**
**(A)** Schematic of the CD8α (aa 1–206)-(HTR6-CT)-BioID2-HA constructs used here, showing only the HTR6-CT moiety. The CTS2 is shown as a blue oval and red crosses indicate the CT-mut3 mutation. Dashed lines indicate deleted regions. The intensity of TULP3 association is displayed on the right. At bottom, the regions in HTR6-CT promoting TULP3 association are shown in green, and the CTS2-containing region antagonizing TULP3 association is shown in red. **(B)** SDS–PAGE and WB analysis of tandem immunoprecipitated S-tagged TULP3 (top two panels) and of cleared cell lysates (bottom two). In the IPs, NeutrAvidin-HRP was used to detect TULP3 biotinylation (top) and anti-Stag antibody to detect its total levels. In the lysates, anti-EGFP and anti-HA tag antibodies were used to detect EGFP-TEV-Stag-TULP3 and CD8 fusions, respectively. Molecular weight markers on the right (kD). **(B, C)** Quantitation from (B) of biotinylated TULP3 signal, relative to total TULP3 in IP, and normalized to β2AR-CT sample. Data are mean ± SEM (n = 7, 6, 7, 5, 5, 3, 5, 4, 4 independent experiments, from left to right). Data were analyzed by one-way ANOVA followed by Tukey's multiple comparisons tests. Significance shown as $P < 0.05$(*), $P < 0.001$(***), or $P < 0.0001$(****). Where not explicitly indicated, asterisks represent significance relative to β2AR-CT. Source data are available for this figure.

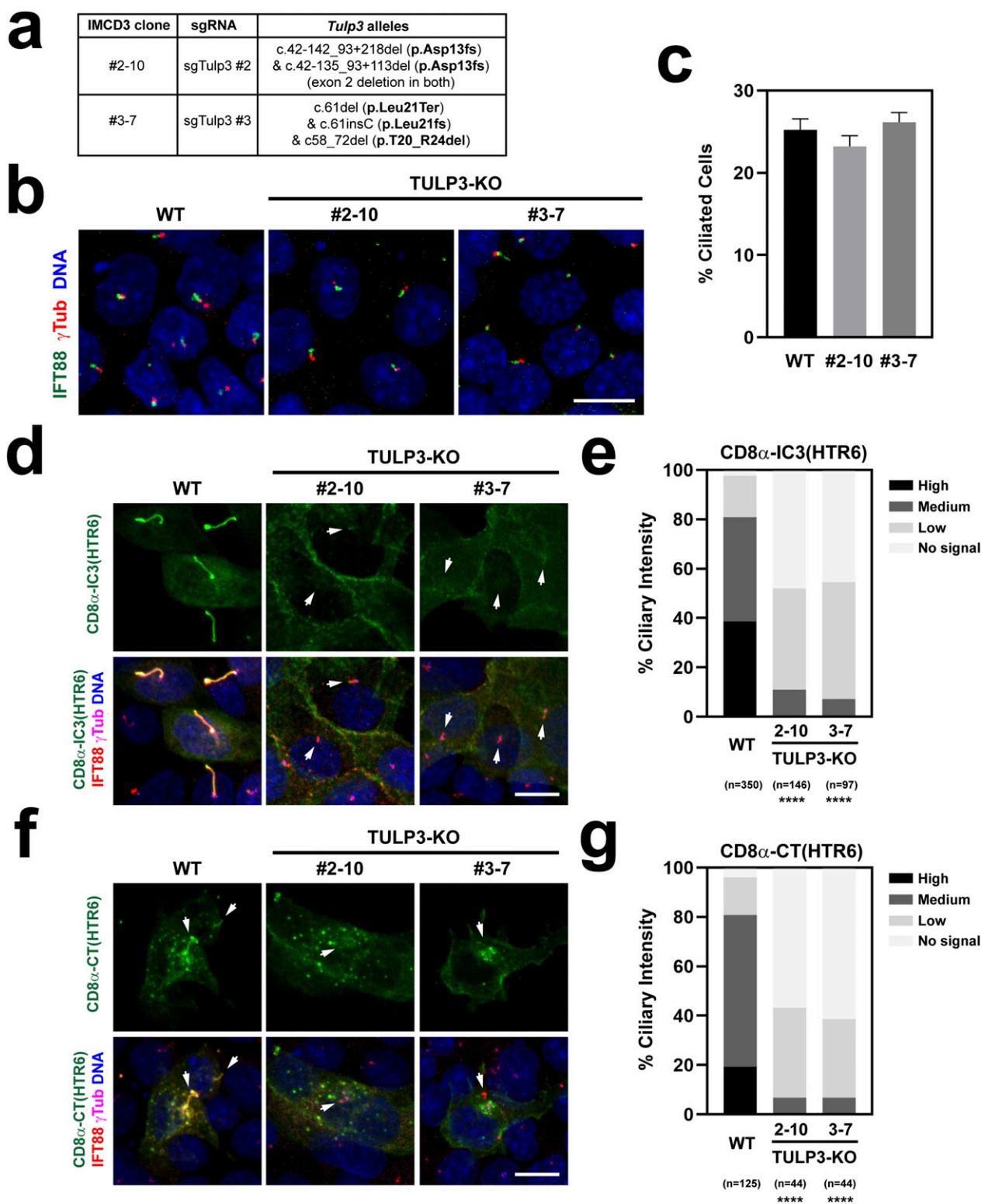

**Figure 11. TULP3 regulates both CTS1 and CTS2 function.**
**(A)** Genomic characterization of the two CRISPR-generated TULP3-KO IMCD3 clones used in this figure. Allele nomenclature corresponds to Ensembl mouse transcript Tulp3-201 and follows Human Genome Variation Society (HGVS) guidelines (52). **(B)** TULP3-KO clones still form cilia, as seen by immunostaining with IFT88 (green) and g-tubulin (red) antibodies. DAPI in blue. Scale bar, 10 μm. **(B, C)** Percentage of ciliated cells was quantitated from (B). Data are mean ± SEM of n = 33 fields of cells, each containing at least 30 cells, from two coverslips. No significant differences were found by one-way ANOVA. **(D)** CD8α-IC3(HTR6)-Stag-TEV-EYFP was transfected into TULP3-KO clones, or WT IMCD3 cells as control, and its ciliary localization assessed by immunostaining with Stag (green), IFT88 (red), and g-Tubulin (magenta) antibodies.

expression levels of their respective CD8α fusions (Fig 12C), an effect that was not observed in our full length myc-HTR6 experiment (Fig 12B). After normalizing data to account for reduced expression, IC3-mut4 still reduced RABL2B binding, whereas CT-mut3 increased it (Fig 12C). Altogether, our data suggest that RABL2B binds HTR6 through both HTR6-IC3 and HTR6-CT. Moreover, whereas CTS1 is important for RABL2B binding to HTR6-IC3, CTS2 is dispensable for RABL2B binding to HTR6-CT.

### RABL2 promotes HTR6 ciliary targeting mostly via HTR6-IC3

As performed for TULP3, we then explored how RABL2 regulates CTS1 and CTS2 function. Using CRISPR in mouse IMCD3 cells, we obtained two RABL2-KO clones, #1-4 and #3-12 (Fig 12D). Consistent with previous reports, ciliogenesis in these clones was clearly reduced, but not completely suppressed (Fig 12E and F). We next assessed ciliary targeting of EYFP-tagged CD8α-(HTR6-IC3) or CD8α-(HTR6-CT) in these clones as compared to WT cells (Fig 12G–J). For CD8α-(HTR6-IC3), ciliary targeting was strongly reduced in both RABL2-KO clones, in which ciliary intensity was consistently low or undetectable, as opposed to medium–high in WT cells (Fig 12G and H). For CD8α-(HTR6-CT), ciliary targeting was also significantly reduced, but less markedly so (Fig 12I and J). These data indicate that RABL2 affects both CTS1 and CTS2-dependent HTR6 ciliary targeting, but is especially important for CTS1-dependent targeting through HTR6-IC3.

### Effect of TULP3 on RABL2-HTR6 binding

Last, we began to explore the interplay between TULP3 and RABL2 in HTR6 ciliary targeting. To address this, we tested whether TULP3-myc affects how EGFP-RABL2B co-IPs HTR6-Flag (Fig 12K). Coexpression of wild-type TULP3-myc (or of myc-B9D1, a negative control) did not affect total levels of RABL2B-bound HTR6. However, reduced HTR6-Flag co-IP was seen with TULP3-KR-myc (K268A+-R270A), a mutant abolishing TULP3's phosphoinositide-binding ability, and perhaps also its acetylation (Fig 12K) (30, 40). Moreover, for TULP3-WT-myc, and even more so for TULP3-KR-myc, we observed moderately reduced co-IP of HTR6-Flag's lowest molecular weight band (roughly corresponding to its predicted unmodified size of 48.7 kD). However, this band's intensity varies in different experiments and with different antibodies, so the significance of this remains unclear (Figs 12A, B, and K and S7). In any event, these data suggest that TULP3-dependent regulation of HTR6-RABL2 interactions is an area worthy of further exploration.

## Discussion

Herein, we made several important contributions to our understanding of HTR6 and SSTR3 ciliary targeting. We discovered that

both GPCRs contain CTSs not only in IC3 but also in CT, and that these CTSs act redundantly to promote ciliary targeting in both IMCD3 cells and neurons. We mapped CTS1 and CTS2 in both GPCRs, thereby identifying novel ciliary targeting motifs and shedding light on the role of the previously reported A-Q motifs. We also characterized how these CTSs interact with TULP3 and RABL2, and how these ciliary trafficking adapters regulate function of these CTSs (Table 1).

Initially, functional redundancy between CTS1 and CTS2 appeared partial, as both Chimera D (lacking CTS2) and Chimera J (lacking CTS1) localize to cilia less efficiently than wild-type HTR6 (Fig 3). Nevertheless, HTR6(CT-mut3), wherein CTS2 is specifically disrupted by point mutations (LLL398-400AAA), localizes to cilia as efficiently as wild-type HTR6 (Fig 5), indicating that CTS2 is fully redundant. The fact that CTS1 and CTS2 both drive ciliary accumulation of CD8α chimeras with comparably high efficiency further suggests redundancy is nearly or fully complete (Fig 6D–F). If so, introducing IC3-mut4 (RKQ216-218AAA) into wild-type HTR6 should not significantly reduce ciliary targeting. Such strong redundancy suggests that loss of ciliary targeting of these receptors is deleterious and evolutionarily selected against. Consistently, mounting evidence indicates that ciliary targeting is critically important for the function of ciliary GPCRs such as HTR6 (5, 6, 9, 13, 19).

We also shed light on the role of A-Q motifs. As previously reported (25), we confirmed that Chimera N, which contains the first half of HTR6-IC3 in an HTR7 background, localizes to cilia, whereas Chimera N (A230F+Q234F), lacking the canonical 230-ATAGQ-234 motif, fails to do so (Fig 2A–C). However, we also demonstrated that this AQ>FF mutation strongly impairs Chimera N's plasma membrane targeting, likely explaining why it does not reach cilia (Fig 2D and E). Furthermore, ciliary targeting of HTR6(CT-mut3) is unperturbed by the same AQ>FF mutation, or by the IC3-Δ3 deletion removing the whole A-Q motif (Fig 5A–F). Altogether, this shows that the A-Q motif in HTR6-IC3 is not required for CTS1 function. Accordingly, a canonical Ax[AS]xQ motif is only present in murine HTR6 but not in humans or other mammals, whereas the RKQ motif we identified is conserved across vertebrate HTR6 orthologs (Fig S8). Intriguingly, the RKQ motif is preceded by two alanines, making it a non-canonical AxxxQ motif (214-AARKQ-218) (24, 25). The significance of this is unclear, however, as R216 matters more than Q218 for HTR6 ciliary targeting (Fig 5G and H). On the other hand, alanines 214–215 are fully conserved in vertebrates, but then so are leucines 212–213 and other nearby residues dispensable for CTS1 function (Figs S8A and 5A, E, and F). Checking how A214F and A215F mutations affect HTR6 intracellular and ciliary trafficking would clarify this issue.

The story is different for SSTR3, which has two canonical AP[AS]CQ motifs in rodents, but only one in humans (Fig S8C). Mutating both of these A-Q motifs to F-F motifs fully abolishes CTS1 function without affecting plasma membrane trafficking (Figs 7D–H and S5).

---

DAPI-stained nuclei in blue. CD8α-IC3(HTR6) levels are very low or undetectable in TULP3-KO cilia (arrows). Scale bar, 10 μm. **(D, E)** Quantitation of CD8α-IC3(HTR6) ciliary intensity from (D). The percentage of cilia in each of the indicated categories is shown. The number of transfected cell cilia counted in each condition is displayed at the bottom, together with statistical significance from chi-square tests comparing each mutant distribution to that of WT (P < 0.0001 (****)). **(D, F)** Same analysis as in (D) was performed for CD8α-CT(HTR6)-Stag-TEV-EYFP, which again localizes to WT but very weakly or not at all in TULP3-KO cilia (arrows). **(E, F, G)** Quantitation of CD8α-CT(HTR6) ciliary intensity from (F) was performed and analyzed as in (E).

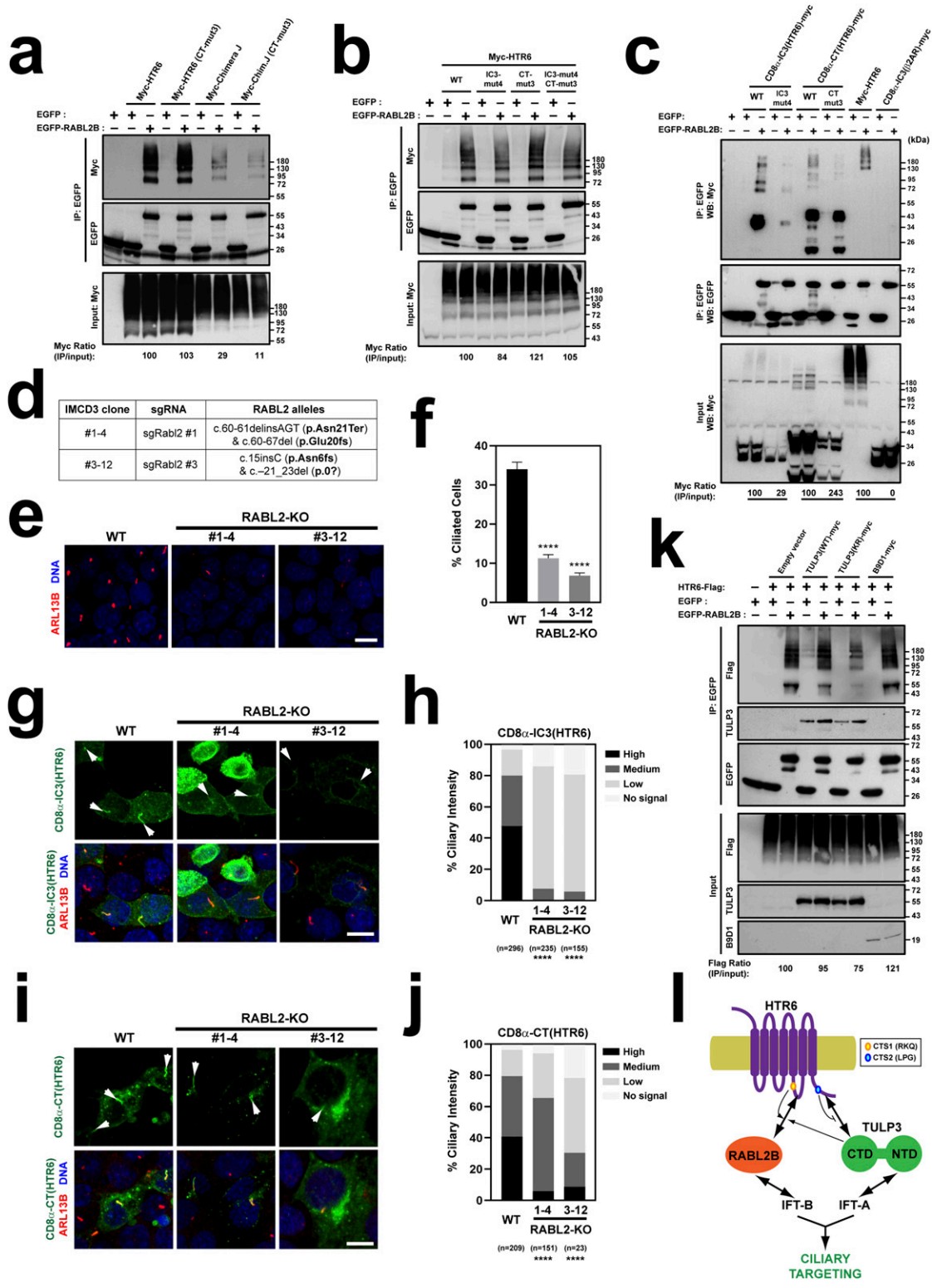

**Figure 12. RABL2 regulates HTR6 ciliary targeting mostly via CTS1.**
**(A)** Lysates from HEK293T cells expressing the proteins indicated at the top were immunoprecipitated with anti-EGFP antibodies and analyzed by Western blot with anti-Myc and anti-EGFP antibodies, as indicated. At bottom, Myc signal immunoprecipitated by EGFP-RABL2 is quantitated relative to Myc signal in the corresponding lysate, and normalized relative to Myc-HTR6 (100%). Molecular weight markers are shown on the right. Input is 2.5% of lysate used for IP. **(A, B)** Co-IP experiment as in (A), but with the constructs indicated at the top. **(A, C)** Co-IP experiment as in (A), but with the constructs indicated at the top. In the quantitations below, each mutant is normalized to its respective control (and β2AR-IC3 is normalized to HTR6). **(D)** Genomic characterization of the two CRISPR-generated RABL2-KO IMCD3 clones used in this figure. Allele

However, this is partly due to dominant effects of the added phenylalanines, as concomitant deletion of both A-Q motifs only partially reduces CTS1 function, again without interfering with cell surface expression (Figs 7D–K and S5). Still, this partial reduction, which affects ciliary intensity more strongly than cilia positivity (Fig 7I and J), indicates that the A-Q motifs in SSTR3 play an important and specific role in ciliary targeting. Because no impairment in CTS1 function is seen by separately deleting each of the two A-Q motifs in mouse SSTR3, these motifs probably act redundantly (Fig 7I and J). Whether the single A-Q motif in human SSTR3 is needed for CTS1 function remains untested.

Which residues within the tandem AP[AS]CQ motifs affect CTS1 function of SSTR3-IC3 also remains unclear. The cysteines are good candidates, as they are required for SSTR3-IC3 to bind the BBSome, a ciliary cargo adapter required for SSTR3 ciliary accumulation (36, 41). Furthermore, these cysteines are needed for proper ciliary targeting of CD8α-(SSTR3-IC3) (36). Still, the roles of these cysteines have not yet been tested in full length SSTR3. This can now be done by mutating them in combination with a CTS2-disrupting mutation. The same applies to the conserved alanines, prolines, and glutamines, which may also have important roles in CTS1 function (25, 28, 36).

In HTR6, we fully mapped the individual residues required for CTS1 and CTS2 function, leading to identification of RKQxxxV and LPG motifs in IC3 and CT, respectively. The former is highly conserved in vertebrates, whereas the latter is restricted to mammals (Fig S8A and B). Beyond HTR6 orthologs, most known ciliary GPCRs do not have these exact same motifs (not shown). Still, there are exceptions (e.g., GPR88, GPR161, GPR63, and PTGER4 all have LPG motifs in their CTs (34, 42)), and most ciliary GPCRs do have similar motifs that may potentially perform the same function (e.g., many have RK or similar motifs in their IC3s). However, for these analyses to be more meaningful, we would need to know which residue substitutions preserve or disrupt CTS function at each position within the motif. Thus, these analyses are somewhat premature at this point. As increasing numbers of GPCR CTSs are discovered and characterized, in silico approaches to predict such CTSs should eventually become more reliable.

Our mapping of CTS1 and CTS2 in SSTR3 was less exhaustive than for HTR6, but still very informative. CTS1 function of SSTR3-IC3 relies on the already discussed A-Q motifs, and on an arginine-rich stretch immediately thereafter (256-RRRRSERR-263) (Fig 7I and J). This stretch is very conserved in vertebrates (Fig S8C), but its effect on ciliary targeting is at least partly due to its role in plasma membrane trafficking, as its deletion causes a marked increase in intracellular retention (Figs 7K and S5). Higher resolution mutagenic analysis of this stretch may

potentially uncouple its effects on cell surface expression from any specific ciliary targeting roles. If such specific roles exist, this would mean that both HTR6-IC3 and SSTR3-IC3 rely on basic residues for CTS1 function, as shown for other ciliary GPCRs such as NPY2R (26).

In SSTR3-CT, we identified the juxtamembrane region as critical for CTS2 function. In this region, an essential role is played by the FK motif, which is required for BBSome binding and is homologous to Smoothened CTS (29, 43). Also critical is the LLxP motif, which vaguely resembles HTR6's LPG and Rhodopsin's VxP (22, 24). Whereas the FK motif is conserved throughout vertebrates, LLxP is restricted to mammals (Fig S8D). Nevertheless, the significance of species conservation is unclear, as it is not yet known whether SSTR3 (or HTR6) localizes to non-mammalian cilia. Beyond the juxtamembrane region, deleting SSTR3-CT's glutamate-rich coiled coil reduces cilia localization by half (Figs 7D–G), prompting the speculative hypothesis that ciliary accumulation of SSTR3 may be reinforced by electrostatic interactions between its glutamate-rich C-terminal coiled coil and its arginine-rich IC3.

We also shed light on the mechanisms of action of these CTSs. We established that HTR6 ciliary targeting is TULP3-dependent (Fig 1), as shown previously for SSTR3 and other ciliary GPCRs (27, 30, 34). Moreover, we showed that TULP3 is important for both CTS1 and CTS2 function in HTR6 (Fig 11). TULP3 acts as a ciliary trafficking adapter by connecting ciliary membrane cargo to the IFT machinery. Interaction with the latter is dependent on TULP3's N-terminal domain, whereas cargo association relies on its C-terminal Tubby domain (30, 34). Accordingly, we found that HTR6 associates with TULP3 via the Tubby domain (Fig S6).

We also discovered that TULP3 association to ciliary GPCRs can be mediated by either IC3 (as previously shown for MCHR1 and GPR161 (34), and confirmed herein for the latter [Fig 9]) or CT (as shown here for HTR6 and SSTR3 [Fig 9]). Furthermore, we showed that TULP3 binding to HTR6-CT does not involve the key CTS2 residues, which antagonize rather than promote this interaction (Fig 10). This suggests that TULP3 dissociation is a critical step for HTR6 ciliary accumulation. This hypothesis is consistent with other evidence pointing to the following multistep model of TULP3-mediated ciliary GPCR trafficking: (1) TULP3 associates with cargo GPCRs at the $PI(4,5)P_2$-rich ciliary base; (2) TULP3 binding to IFT trains allows it to ferry cargo across the TZ and into the ciliary membrane; (3) low $PI(4,5)P_2$ levels at the ciliary membrane prompt TULP3-cargo dissociation; and (4) TULP3 is recycled back across the TZ to the ciliary base, where it can engage in new cycles of cargo transport (30, 34, 44, 45, 46, 47). According to this model, failure to

nomenclature corresponds to Ensembl mouse transcript Rabl2-201 and follows Human Genome Variation Society (HGVS) guidelines (52). **(E)** Ciliogenesis is strongly reduced in RABL2-KO clones, as seen by immunostaining with ARL13B (red) antibodies. DAPI in blue. Scale bar, 10 $\mu$m. **(E, F)** Percentage of ciliated cells was quantitated from (E). Data are mean ± SEM of n = 12 fields of cells, each containing at least 30 cells, from two coverslips. Data were analyzed by one-way ANOVA with Tukey's multiple comparisons tests ($P < 0.0001$ (****)). **(G)** EYFP-tagged CD8α-IC3(HTR6) was transfected into RABL2-KO clones, or WT IMCD3 cells as control, and its ciliary localization assessed by immunostaining with EGFP (green) and ARL13B (red) antibodies. DAPI-stained nuclei in blue. CD8α-IC3(HTR6) levels are very low or undetectable in RABL2-KO cilia (arrows). Scale bar, 10 $\mu$m. **(G, H)** Quantitation of CD8α-IC3(HTR6) ciliary intensity from (G). Percentage of cilia in each of the indicated categories is shown. Number of transfected cell cilia counted in each condition is displayed at the bottom, together with statistical significance from chi-square tests comparing each mutant distribution to that of WT ($P < 0.0001$ (****)). **(G, I)** Same analysis as in (G) was performed for CD8α-CT(HTR6)-EYFP. Arrows point to cilia. Scale bar, 10 $\mu$m. **(H, I, J)** Quantitation of CD8α-CT(HTR6) ciliary intensity from (I) was performed and analyzed as in (H). **(A, K)** Co-IP experiment as in (A) but with the indicated constructs and antibodies. **(L)** Model of HTR6 ciliary targeting. Double-headed arrows represent physical interactions. Single-headed arrows represent positive effects. Also shown is the antagonism of CTS2 on TULP3 binding to HTR6-CT. We hypothesize CTS2 promotes intraciliary dissociation of TULP3, thereby freeing it for further rounds of transport. Whether HTR6-IC3 directly binds to IFT-A, as shown for SSTR3-IC3, remains unknown (48).
Source data are available for this figure.

**Table 1.** Summary of key results.

| | | Sufficiency for cilia targeting (Figs 3, 6, and 7) | Residues required for cilia targeting (Figs 4, 5, and 7) | TULP3 | | | RABL2 | | |
|---|---|---|---|---|---|---|---|---|---|
| | | | | IC3/CT association (Fig 9) | CTS effect on association[a] (Fig 10) | Requirement for CTS function (Fig 11) | IC3/CT association (Fig 12A–C) | CTS effect on association[b] (Fig 12A–C) | Requirement for CTS function (Fig 12G–J) |
| HTR6 | IC3 | + | RKQ...V (aa 216–222) | – | n.a. | Strong | ++ | Positive | Strong |
| | CT | + | LPG (aa 400–402) | + | Negative | Strong | ++ | –/+ | Weak |
| SSTR3 | IC3 | + | AP[AS]CQ (aa 243–255) RRRRSERR (aa 256–263) | – | n.a. | ? | ? | ? | ? |
| | CT | + | FK (aa 329–330) LL.P (aa 337–340) | +++ | ? | ? | ? | ? | ? |

[a]n.a., not applicable.
[b]Effects of mutating CTS2 in HTR6-CT were context-dependent and not obvious (see Fig 12A–C for more details).

release TULP3 inside the ciliary membrane would prevent TULP3 recycling and its catalytic action as a ciliary transporter. This could explain the loss of HTR6 ciliary targeting we see with mutations causing excessive TULP3 binding.

Logically, if TULP3 dissociation is required for ciliary GPCR targeting, then previous TULP3 association must also be required. This was already proven for GPR161, whose IC3 CTS is needed for TULP3 binding (34). For HTR6, we found that different regions in its CT, both before and after CTS2, promote TULP3 association, including residues 373–389, whose deletion from HTR6-CT completely abolishes the interaction (Fig 10). Thus, one might expect these residues to be needed for HTR6 ciliary accumulation. Although we did not directly check this, we saw that deleting residues 371–378 or 379–391 does not impede targeting of Chimera J, in which HTR7-IC3 replaces HTR6-IC3 (Fig 4). However, these two Chimera J deletions may not suffice to abolish HTR6-CT binding to TULP3.

Alternatively, HTR6 may also bind TULP3 via sequences other than CT. Although we found no association between TULP3 and HTR6-IC3 in our proximity biotinylation experiments, we saw a clear effect of TULP3 on HTR6-IC3 ciliary targeting (Figs 9 and 11). The reason for this is unclear, but we do not rule out a weak or transient interaction between TULP3 and HTR6-IC3, or an interaction that does not lead to efficient TULP3 biotinylation by CD8α-(HTR6-IC3)-BioID2. Interestingly, SSTR3-IC3 interacts directly with the IFT–A complex, which binds TULP3, and yet we found no TULP3 biotinylation by CD8α-(SSTR3-IC3)-BioID2 either (Fig 9) (30, 48). This may mean that IFT-A interactions with TULP3 and SSTR3-IC3 are mutually incompatible, or that TULP3-IFT-A binds SSTR3-IC3 in a way that keeps TULP3 away from BioID2's active site. More work is required to clarify this, as well as whether TULP3 binds HTR6 and/or SSTR3 through regions other than IC3 and CT.

Like TULP3, RABL2 functions as an adapter connecting IFT trains to ciliary GPCRs. And like TULP3, RABL2 is required for targeting ciliary GPCRs such as HTR6 and GPR161 (31). Recently, RABL2 has also been implicated in BBSome-dependent ciliary GPCR exit (49). Our binding experiments indicated that RABL2 interacts with both HTR6-IC3 and HTR6-CT, and that binding to HTR6-IC3 is dependent on CTS1 residues, whereas binding to HTR6-CT does not require CTS2 residues

(Figs 12A–C and S7). In fact, as seen for TULP3, CTS2 residues may even antagonize RABL2 binding to HTR6-CT, a point that merits further study (Fig 12C). We also showed that RABL2 is strongly required for CTS1-mediated ciliary targeting, but less important for CTS2 function (Fig 12D–J). Our data also suggest possible phosphoinositide-dependent effects of TULP3 on HTR6-RABL2 binding (Fig 12K).

Altogether, our data support a model wherein IC3 and CT work together to recruit ciliary trafficking proteins such as TULP3 and RABL2, which connect the GPCR to the IFT machinery for ciliary transport (Fig 12L). After entering cilia, CTS2 may play a prominent role in shedding these trafficking adapters, thereby releasing both GPCRs and adapters so that they can go on with their functions. Although many key players in ciliary GPCR targeting have now been identified, including cis-acting CTSs and trans-acting trafficking proteins, the complex and step-by-step molecular interactions controlling this process remain poorly understood. We believe the advances made herein will aid progress towards a deeper understanding of these mechanisms.

# Materials and Methods

### Reagents and antibodies

Mouse monoclonal antibodies: acetylated Tubulin (T7451, IF: 1: 10,000; Sigma-Aldrich), ARL13B (66739-1-Ig, IF: 1:300; Proteintech, 75-287, IF: 1:1,000; NeuroMab), and S-tag (MAC112, IF: 1:100, WB: 1:5,000; EMD Millipore). Rat monoclonal antibody: HA (7c9, WB: 1:1,000; ChromoTek). Rabbit polyclonal antibodies: EGFP (50403-2-AP, IF: 1:200, WB: 1:1,000; Proteintech), Flag (F7425, WB: 1:1,000; Sigma-Aldrich), Adenylyl cyclase III (C-20, IF: 1:100; Santa Cruz), B9D1 (NBP2-84489, WB: 1:1,000; Novus Biologicals) and HTR6 (31). Goat polyclonal antibodies: γ-Tubulin (sc-7396, IF: 1:200; Santa Cruz). Alexa Fluor (AF)–conjugated donkey secondary antibodies from Thermo Fisher Scientific (all used for IF at 1: 10,000): AF488 anti-mouse IgG (A21202), AF488 anti-rabbit IgG (A21206), AF555 anti-mouse IgG (A31570), AF555 anti-rabbit IgG (A31572), AF594 anti-rabbit IgG (A21207), and AF647 anti-goat IgG (A21447). Also from Thermo Fisher Scientific were HRP-conjugated secondary antibodies

(used for WB at 62 ng/ml): goat anti-rabbit IgG (A16104), goat anti-mouse IgG (A16072) and donkey anti-rat IgG (A18739). Biotinylated proteins were detected with Neutravidin-HRP (A2664, 1 µg/ml; Thermo Fisher Scientific). D-biotin was from Thermo Fisher Scientific (BP-232-1).

## Plasmids

Details of all plasmids used in this study can be found in Table S1. Htr6-EGFP, Htr7-EGFP, Sstr3-EGFP, Sstr5-EGFP, Htr7[TM5-V241Htr6]-EGFP (Chimera N), and the latter's AQ>FF mutant have been described (25), as have EGFP-TEV-Stag-TULP3, EGFP-RABL2B, and Flag-RABL2B (31, 34). Chimeric constructs, internal deletions and missense mutations were generated by overlap extension PCR (25). Most other constructs were created by PCR-amplifying the region of interest using primers containing restriction enzyme targets, and mutations where needed. Amplifications were performed with Platinum SuperFi DNA Polymerase (Thermo Fisher Scientific), and the resulting PCR products were digested and ligated into desired vectors. Sequences of all plasmids were confirmed by Sanger DNA sequencing (Eurofins Genomics). Primer sequences and PCR conditions are available on request.

## Cell culture and transfection

Murine inner medullary collecting duct 3 (IMCD3) cells, and their HTR6-IMCD3 derivative clone (31), were cultured in DMEM/F12 medium supplemented with 10% FBS. Human embryonic kidney 293T (HEK293T) cells were maintained in DMEM with 10% FBS. All cell lines were grown at 37°C and 5% $CO_2$ in a humidified atmosphere and were mycoplasma-free, as ascertained by regular tests. IMCD3 cells were reverse transfected at ~50% confluency using JetPrime (Polyplus-transfection) and their cilia analyzed 65 h later, without serum starvation. RNAi experiments with HTR6-IMCD3 cells are described in next section. HEK293T cells were transfected using either PEI Max (Polysciences) or the calcium phosphate method. Primary hippocampal neurons were cultured as previously described and transfected using Lipofectamine LTX & Plus Reagent methods (Life Technologies/Invitrogen) 7 d after plating (50).

## RNA interference

For RNAi, $1 \times 10^5$ HTR6-IMCD3 cells were seeded in 24-well plates, cultured for 24 h and transfected with 20 pmol siRNA using Lipofectamine RNAiMAX (Thermo Fisher Scientific). Transfected cells were cultured in normal medium for 24 h and then serum-starved for 48 h before analysis. siRNA oligonucleotides (Sigma-Aldrich) were siLuc (5'-CGUACGCGGAAUACUUCGAUU-3'), siTulp3 #1 (5'-GAAA-CAAACGUACUUGGAUtt-3'), and siTulp3 #2 (5'-GCAGCUAGAAAGCGGA-AAAtt-3').

## Generation of CRISPR cell lines

Three single guide RNAs (sgRNAs) were designed for mouse *Tulp3* and *Rabl2* genes using the CRISPOR web tool (51). Oligonucleotides encoding these sgRNAs were annealed and inserted into the pSpCas9(BB)-2A-Puro (PX459) V2.0 vector (#62988; Addgene) using BbsI restriction sites. sgRNA sequences were AGATGAT-CTTCACGTTCTCA (sgRabl2#1), CGTGAAGATCATCTGCCTGG (sgRabl2#2),

CGATGGCAGGGGACAGAAAC (sgRabl2#3), CACCGGTCGGGCGGGTATGG (sgTulp3#1), CCTCAGGGTCTCATCGTCAA (sgTulp3#2), and CCTTTGAC-GATGAGACCCTG (sgTulp3#3). JetPrime (Polyplus-transfection) was then used to reverse transfect 1,200 ng of the sgRNA and Cas9-encoding plasmids into IMCD3 cells at ~80% confluency on six-well plates. 24 h post-transfection (hpt), puromycin (2 µg/ml) was added to the cell medium for 72 h. At 96 hpt, puromycin-resistant cells were transferred to 15-cm dishes and allowed to grow (in puromycin-free medium from here on) until cell colonies were visible. Cloning cylinders were then used to transfer individual colonies to separate wells. After expansion, these clones were analyzed by IF with ciliary markers to identify likely TULP3-KO and RABL2-KO clones (unfortunately, due to technical issues, our attempts to directly visualize TULP3 and RABL2 loss by WB or IF were unsuccessful). For each gene, two likely mutant clones were selected for genomic analysis. The region of interest was amplified with Platinum SuperFi DNA Polymerase (Thermo Fisher Scientific) and cloned into pJET1.2/blunt using the CloneJET PCR cloning kit (Thermo Fisher Scientific). For each IMCD3 clone, plasmid DNA from 10 bacterial colonies was Sanger sequenced (Eurofins Genomics), leading to identification of the alleles shown in Figs 11A and 12D.

## Quantitative PCR

Total RNA was isolated from cultured cells using Sepasol (Nacalai Tesque) and reverse transcribed with ReverTra Ace qPCR RT kit (Toyobo). Quantitative PCR was performed using Thunderbird SYBR qPCR mix (Toyobo) and LightCycler96 (Roche). Data were analyzed with ΔΔCt method using siLuc and Gapdh as controls. Primers: Tulp3_F (5'-CAGCTGAAGCTGGACAATCA-3'), Tulp3_R (5'-GGGTTTGGCTGTAC-CATGAG-3'), Gapdh_F (5'-AACTTTGGCATTGTGGAAGG-3'), and Gapdh_R (5'-TGCAGGGATGATGTTCTGG-3').

## Immunofluorescence and quantitations

IMCD3 cells were grown on coverslips, fixed 5 min at RT in PBS + 4% PFA followed by 3 min at −20°C in freezer-cold methanol. Cells were then incubated 30–60 min at RT in blocking solution (PBS + 0.1% Triton X100+2% donkey serum + 0.02% sodium azide), which was also used to dilute primary antibodies. 1 µg/ml DAPI (Thermo Fisher Scientific) was added with secondary antibodies, which were diluted in PBS. After the last round of PBS washes, coverslips were mounted on slides using Prolong Diamond (Thermo Fisher Scientific), incubated overnight at 4°C and imaged with a Leica TCS SP5 confocal microscope. For HTR6-IMCD3, cells were fixed 10 min at RT in PBS + 3.7% formalin, permeabilized 10 min in PBS + 0.2% Triton X-100 and blocked and stained in PBS + 5% BSA. Hoechst33342 (Nacalai Tesque) was used as DNA stain and PermaFluor (Thermo Fisher Scientific) as the mounting medium. Cells were imaged using a Zeiss Axi-oObserver microscope. Primary hippocampal neurons on coverslips were fixed 24 h post-transfection in 4% PFA + 10% sucrose (15 min, RT), permeabilized in PBS + 0.3% Triton X-100+4% donkey serum + 1% BSA + 0.02% sodium azide (10 min, RT), incubated with anti-ADCY3 antibodies (16–24 h, 4°C), washed thrice in PBS + 4% donkey serum + 1% BSA + 0.02% sodium azide (5 min each),

incubated 1 h in Alexa Fluor 546-conjugated goat anti-rabbit IgG (1 h, RT), washed again and mounted using Immuno-Mount (Thermo Fisher Scientific). Nuclei were visualized with DRAQ5. Samples were imaged on a Leica TCS SP8 laser scanning confocal microscope at the Hunt-Curtis Imaging Facility in the Department of Neuroscience at The Ohio State University. Brightness and contrast of microscopic images were adjusted for optimal visualization with Leica LAS X, Adobe Photoshop, or Fiji (Image J). To quantitate percentage of positive cilia for the proteins of interest (GPCRs, chimeras, or CD8α fusions), cells that were both transfected and ciliated were counted in seven representative fields spanning the entire coverslip. At least 50 such cells were counted per coverslip and experiment. Cells with no signs of cell surface expression were not counted (such cells were very rare, less than 1%, unless otherwise noted in the text). Quantification of HTR6 intensity in RNAi experiments was performed with Image J as described ([53]). To quantify ciliary signal intensity of SSTR3 mutants, at least 25 cilia per condition were imaged from a representative experiment. Imaging parameters were constant across samples and chosen to avoid signal saturation. Quantitation was then performed on 12-bit images using Fiji to measure average pixel intensity within each cilium and subtracting from it average pixel intensity of an equally sized background region. To quantitate intracellular retention, EGFP-tagged GPCR-transfected cells displaying no plasma membrane or ciliary membrane staining were counted relative to total number of transfected cells.

### Proximity biotinylation assays

Proximity biotinylation experiments were based on reference 34, with some modifications. Briefly, HEK293T cells were cotransfected with plasmids encoding EGFP-TEV-Stag-TULP3 and the proteins of interest fused to BioID2, an improved and smaller biotin ligase ([37]). Transfected cells were grown in presence of 50 $\mu$M D-biotin for the last 16 h before cell lysis. Cells were harvested 48 h post-transfection and incubated in lysis buffer (50 mM Tris–HCl, pH 7.4, 200 mM KCl, 1 mM MgCl$_2$, 1 mM EGTA, 10% glycerol, 1 mM DTT, 0.6% IGEPAL CA-630, and Halt protease inhibitor cocktail [#78429; Thermo Fisher Scientific]) at 4°C for 30 min, with rotation. Lysates were then cleared by centrifugation at 4°C for 10 min at 20,000$g$. Tulp3 was purified from these lysates by tandem affinity purification. First, EGFP-TEV-Stag-TULP3 was immunoprecipitated with GFP-Trap_MA beads (ChromoTek) for 2 h at 4°C, with rotation. Immunoprecipitates were then digested for 1 h at 25°C with TEV protease (#7847; BioVision) in a buffer containing 50 mM Tris–HCl, pH 8, 100 mM NaCl, and 5 mM DTT, according to manufacturer's instructions. Stag-TULP3 was then pulled down from the TEV digestion eluates by overnight 4°C rotational incubation using S-protein agarose beads (#69704; Merck). Beads were then eluted with Laemmli buffer and processed for SDS–PAGE in Novex Value 4–20% Tris-Glycine gels (Thermo Fisher Scientific). Proteins were then transferred to nitrocellulose or polyvinylidene fluoride (PVDF) membranes and analyzed by immunoblot. For electrochemiluminescent detection, X-ray film or an ImageQuant LAS 500 chemiluminescence CCD camera (GE Life Sciences) were used. Immunoblots were quantified using Fiji/Image J.

### Immunoprecipitation and Western blot

For EGFP-RABL2B immunoprecipitations, cells were harvested 48 h post-transfection and lysed in buffer containing 50 mM Tris–HCl pH 7.5, 150 mM NaCl, 0.5 mM EDTA, 1% Igepal CA-630, and Halt protease inhibitor cocktail (#78429; Thermo Fisher Scientific). After rocking 30 min at 4°C in lysis buffer, lysates were cleared by centrifugation at 20,000$g$ and 4°C for 10 min. For immunoprecipitation with GFP-Trap_MA beads (ChromoTek), cleared lysates were rocked for 2 h at 4°C. Beads were then eluted with Laemmli buffer and processed for SDS–PAGE in Novex Value 4–20% Tris-Glycine gels (Thermo Fisher Scientific). Proteins were then transferred to nitrocellulose or PVDF membranes and analyzed by immunoblot. For electrochemiluminescent detection, X-ray film or a Fusion Solo S imaging system (Vilber) were used. For Flag-RABL2B immunoprecipitation, cells were lysed at 4°C for 30 min in lysis buffer (50 mM Hepes pH 7.5, 150 mM NaCl, 5 mM MgCl$_2$, 0.5% NP-40, 1 mM DTT, 0.5 mM PMSF, and 2 $\mu$g/ml leupeptin). Lysates were then cleared by centrifugation (17,000$g$ for 10 min) and their protein content analyzed. Immunoprecipitations were performed on 2 mg of total protein by incubating 2 h at 4°C with anti-Flag agarose beads (Sigma-Aldrich). Beads were then washed in lysis buffer and bound polypeptides analyzed by SDS–PAGE and immunoblotting. 20 $\mu$g of lysate was loaded in the input lane.

### Statistical analysis

GraphPad Prism 8 software was used to graph and statistically analyze data. Specific details of each experiment are provided in the corresponding figure legends.

# Supplementary Information

# Acknowledgements

We thank members of the Garcia-Gonzalo, Kobayashi, and Mykytyn labs for useful discussions. This work was supported by grants from the Spanish Ministry of Science and Innovation (MICINN) to FR Garcia-Gonzalo (PID2019-104941RB-I00, SAF2015-66568-R and RYC2013-14887), the last two cofunded by the European Regional Development Fund (ERDF/FEDER). This work was also supported by research project grant R21 MH121744 from the NIH/NIMH to K Mykytyn, and by Japan Society for the Promotion of Science (JSPS) KAKENHI (No. 18K06627) to T Kobayashi. R Martin-Morales was supported by a MICINN predoctoral grant (BES2016-077828).

## Author Contributions

P Barbeito: conceptualization, investigation, and writing—review and editing.
Y Tachibana: investigation.
R Martin-Morales: investigation.
P Moreno: investigation.

K Mykytyn: conceptualization, investigation, funding acquisition, and writing—review and editing.

T Kobayashi: conceptualization, supervision, funding acquisition, investigation, and writing—review and editing.

FR Garcia-Gonzalo: conceptualization, data curation, supervision, funding acquisition, investigation, writing—original draft, and project administration.

## Conflict of Interest Statement

The authors declare that they have no conflict of interest.

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
