## [Reviewer comments · Life Science Alliance]

Life Science Alliance

HTR6 and SSTR3 ciliary targeting relies on both IC3 loops and C-terminal tails

Pablo Barbeito, Yuki Tachibana, Raquel Martin-Morales, Paula Moreno, Kirk Mykytyn, Tetsuo Kobayashi, and Francesc Garcia-Gonzalo

DOI: <https://doi.org/10.26508/lsa.202000746>

Corresponding author(s): Francesc Garcia-Gonzalo, Instituto de Investigaciones Biomedicas Alberto Sols UAM-CSIC and Pablo Barbeito, Instituto de Investigaciones Biomedicas Alberto Sols UAM-CSIC

Review Timeline:

Submission Date:	2020-04-19
Editorial Decision:	2020-05-28
Revision Received:	2020-11-30
Editorial Decision:	2020-12-11
Revision Received:	2020-12-14
Accepted:	2020-12-15

Scientific Editor: Shachi Bhatt

Transaction Report:

May 28, 2020

Re: Life Science Alliance manuscript #LSA-2020-00746-T

Dr. Francesc R Garcia-Gonzalo
Instituto de Investigaciones Biomedicas Alberto Sols UAM-CSIC
Facultad de Medicina UAM (Lab C-11)
Madrid 28029
Spain

Dear Dr. Garcia-Gonzalo,

Thank you for submitting your manuscript entitled "HTR6 and SSTR3 ciliary targeting relies on both IC3 loops and C-terminal tails" to Life Science Alliance. The manuscript was assessed by expert reviewers, whose comments are appended to this letter.

As you will see, the reviewers appreciate your analyses and provide constructive input on how to further strengthen your work. We would thus like to invite you to submit a revised version of your manuscript to us. The requested changes and additional experimental assays seem rather straightforward to include, but please do get in touch in case you would like to discuss the revision further. Please note that we are aware that many laboratories cannot function fully during the current COVID-19/SARS-CoV-2 pandemic and therefore encourage you to take the time necessary to revise the manuscript to the extent requested. We will extend our 'scooping protection policy' to the full revision period required. If you do see another paper with related content published elsewhere, nonetheless contact us immediately so that we can discuss the best way to proceed.

Thank you for this interesting contribution to Life Science Alliance. We are looking forward to receiving your revised manuscript.

Sincerely,

Andrea Leibfried, PhD
Executive Editor
Life Science Alliance
Meyerhofstr. 1
69117 Heidelberg, Germany
t +49 6221 8891 414
e contact@life-science-alliance.org
www.life-science-alliance.org

B. MANUSCRIPT ORGANIZATION AND FORMATTING:

Reviewer #2 (Comments to the Authors (Required)):

The current manuscript by Barbeito et al characterizes the ciliary targeting sequences (CTS's) of two GPCRs, Htr6 and Sstr3. In doing so, they demonstrate that the CTS's reside in both intracellular loop 3 (IC3) and C-terminal tail (CT) of these GPCRs. The authors nicely show that both these sequences are required for their ciliary targeting. They show previously published Ax[AS]xQ CTS's as partially required (for SSTR3) or not required for ciliary targeting (for Htr6). Importantly, they show that previously published data of HTR6 ciliary localization being affected by mutagenesis of A-Q to F-F results from indirect effects on intracellular accumulation. While both GPCRs require TULP3 for targeting to cilia, they nicely uncouple TULP3 binding from ciliary trafficking. Very interestingly, they show both negative and positive regulatory roles for C-terminal CTS regions in binding to TULP3, and actually a strongly binding mutant to be defective in ciliary targeting. The authors propose that such defects in targeting probably arise from lack of cargo release in cilia. In addition, the authors demonstrate that the recently described RABL2b GTPase that injects IFT-B complexes into cilia, also binds to IC3 of Htr6 and could coordinate trafficking to cilia. Overall, these results explain the redundancy between both CTS's in targeting Htr6 to cilia.

The results are interesting and although I have a few important comments that the authors should try to address to improve their current interpretations, I am overall enthusiastic. That said, the writing and presentation in the present form is at places difficult to read and suffers from too much author-specified abbreviation use. A common table that summarizes the core data regarding important CTS sequences, ciliary localization, TULP3 and Rabl2b binding would hugely improve the presentation for uninitiated readers and generate broad interest.

Major comments:

1. The authors show that both CD8-tagged IC3/CT of Htr6 are ciliary. Are both CTS's TULP3 and/or Rabl2b-dependent, especially as the authors show that CD8-IC3 of Htr6 and Sstr3 do not bind to TULP3?

2. Do the CD8-tagged IC3/CT IC3/CT chimeras bind to Rabl2b? Although the authors show nicely that the IC3 is required in the context of full-length Htr6 for binding to Rabl2, binding in the context of CD8-chimeras would strengthen the idea of direct dependence on Rabl2 binding, particularly for the RKQ motif.

Minor comments:

-Fig 9: Data normalized to IPs?

-Fig 10: Quantification needed. Terms used in describing data such as "moderately reduced", "further reduced" not clear and overtly simplistic without quantification.

-Fig. 10: "CTS2's positive effect is only unveiled in the absence of HTR6-IC3 residues other than those in CTS1": I am not convinced by the authors' claim of the CTS2's effects without quantification.

-Fig S1: Should be included among main figures.

-Htr7 is shown to be 20% positive in cilia. (Fig. 2). Probably a fluorescence intensity plot would better substantiate whether the cilia positive for Htr7 are strongly positive or have poor localization, but nonetheless were quantified as "positive".

Reviewer #3 (Comments to the Authors (Required)):

There is accumulating evidence indicating that primary cilia are important signaling hubs which contain many membrane receptors including G protein-coupled receptors (GPCRs). However, the mechanisms underlying ciliary GPCR targeting remain poorly understood, even though ciliary targeting sequences have been identified in a subset of receptors, such as the serotonin 5-HT₆ receptor and the somatostatin SST3 receptor. The paper of Barbeito et al. that explores molecular determinants of ciliary targeting of both receptors, is an important contribution in the field, as it demonstrates that two ciliary targeting sequences contribute to their ciliary location and provide important new insight into the mechanisms by which these sequences regulate binding to TULP3 and RABL2B, two adapters needed for ciliary GPCR targeting. This study also solves the conundrum that the deletion of the previously identified ciliary targeting sequences in the 5-HT₆ and SST3 receptors does not affect their targeting in the primary cilium while they are sufficient to promote ciliary targeting of non-ciliary receptors.

Overall, this is an outstanding study based on a huge number of mutants and chimera, and the paper is very well written. The experiments are well justified and appear to be well done, the data are logically and clearly presented, and the demonstration is generally convincing and easy to follow. Given the neuronal expression of 5-HT₆ and SST3 receptors, the presence of experiments showing the importance of identified ciliary targeting sequences in primary hippocampal neurons certainly gives additional impact to the paper. I have only a few somewhat minor comments that the authors should take into consideration before acceptance of the manuscript.

1- The data on Fig. 1 would be more convincing if additional controls were provided. The authors show a decrease in TULP3 mRNA in siRNA-treated cells and they are right to do that, but they should also demonstrate the efficiency of silencing TULP3 at the protein level. Most importantly, what is the influence of TULP3 knockdown on 5-HT₆ receptor expression? The data on Fig. 1A show that the 5-HT₆ receptor is no more localized in the primary cilium in siRNA-treated cell. However, no expression of the receptor is shown in other compartments. Larger cell fields as well as Western blots showing TULP3 and 5-HT₆ receptor should be represented.

2- Page 8, bottom: the authors state that both HTR6-CT and SSTR3-CT strongly associated to TULP3, whereas the data on Fig. 8C and 8E show that SSTR3-CT associates much more efficiently to TULP3 than HTR6-CT. Even, HTR6-CT binding to TULP3 does not appear to be significant. Does that mean that 5-HT₆ receptor association with TULP3 mainly relies on its targeting sequence located in IC3?

3- Page 9 and Fig. 10a: The authors conclude from co-immunoprecipitation of myc-5-HT₆ receptor and RABL2B that CTS2 positively regulates the interaction in absence of HTR6-IC3 based on a further reduction of the co-immunoprecipitation when CTS2 is mutated. However, this further reduction is not obvious on the Western blot illustrated on Fig. 10A and the authors should temper this statement in absence on any quantification.

4- The profile of 5-HT₆ receptor immunoreactivity strongly differs from one blot to one another (see for instance Fig. 10A and S7). In some cases, there is a clear signal at the expected molecular weight for monomer (Fig. S7), whereas the corresponding band seems to be absent in other blots (Fig. 10). How do the authors explain that point?

5- In the introduction, page 2, the authors should quote references when they state that 5-HT₆ receptors activate mTOR and Cdk5 signaling. In the same paragraph, regarding the functional outcomes of ciliary targeting of the receptor, they should also quote the paper of Jiang et al. (PNAS 116: 12066-12071, 2019) showing that ciliary localization of 5-HT₆ receptor influences constitutive activation of Gs and cAMP production. This is an important recent work in the field.

Dear Reviewers #2 and #3,

We are very grateful for your time and excellent assessments, which have helped make this work better. Please find our responses to all your points below.

Reviewer #2 (Comments to the Authors (Required)):

The current manuscript by Barbeito et al characterizes the ciliary targeting sequences (CTS's) of two GPCRs, Htr6 and Sstr3. In doing so, they demonstrate that the CTS's reside in both intracellular loop 3 (IC3) and C-terminal tail (CT) of these GPCRs. The authors nicely show that both these sequences are required for their ciliary targeting. They show previously published Ax[AS]xQ CTS's as partially required (for SSTR3) or not required for ciliary targeting (for Htr6). Importantly, they show that previously published data of HTR6 ciliary localization being affected by mutagenesis of A-Q to F-F results from indirect effects on intracellular accumulation. While both GPCRs require TULP3 for targeting to cilia, they nicely uncouple TULP3 binding from ciliary trafficking. Very interestingly, they show both negative and positive regulatory roles for C-terminal CTS regions in binding to TULP3, and actually a strongly binding mutant to be defective in ciliary targeting. The authors propose that such defects in targeting probably arise from lack of cargo release in cilia. In addition, the authors demonstrate that the recently described RABL2b GTPase that injects IFT-B complexes into cilia, also binds to IC3 of Htr6 and could coordinate trafficking to cilia. Overall, these results explain the redundancy between both CTS's in targeting Htr6 to cilia.

The results are interesting and although I have a few important comments that the authors should try to address to improve their current interpretations, I am overall enthusiastic. That said, the writing and presentation in the present form is at places difficult to read and suffers from too much author-specified abbreviation use. A common table that summarizes the core data regarding important CTS sequences, ciliary localization, TULP3 and Rabl2b binding would hugely improve the presentation for uninitiated readers and generate broad interest.

We fully agree that parts of the paper were hard to read. To address this, we have taken several steps:

- (1) Following the reviewer's great suggestion, we have introduced **Table 1**, which summarizes key data from the paper.
- (2) We have made changes to the Discussion section, where it was not really necessary to mention so many specific constructs with their burdensome names. We feel the discussion has improved greatly as a result.
- (3) In the Results section, we have also tried, where possible, to avoid unnecessary complexity.

Regarding author-specified abbreviations (e.g. IC3, CT, CTS1, CTS2), we realize there is a sizable number of them, which we usually avoid. However, given the nature of this work, we feel these abbreviations are warranted, as their omission would make the manuscript even harder to read.

Overall, we think the paper is now significantly better thanks to the reviewer's point.

Major comments:

1. The authors show that both CD8-tagged IC3/CT of Htr6 are ciliary. Are both CTS's TULP3 and/or Rabl2b-dependent, especially as the authors show that CD8-IC3 of Htr6 and Sstr3 do not bind to TULP3?

Great point. To address this, we have used CRISPR to generate TULP3-KO and RABL2-KO clones of IMCD3 cells, as described in methods and results sections, and as shown in **Fig.11a-c** (TULP3-KO) and **Fig.12d-f** (RABL2-KO).

Using these cells, we have found that TULP3 is strongly required for ciliary targeting of both CD8 α -(HTR6-IC3)-EYFP (**Fig.11d-e**) and CD8 α -(HTR6-CT)-EYFP (**Fig.11f-g**), whereas RABL2 is strongly required in the case of CD8 α -(HTR6-IC3)-EYFP (**Fig.12g-h**) but less so for CD8 α -(HTR6-CT)-EYFP (**Fig.12i-j**).

Thus, despite our not detecting TULP3 association with HTR6-IC3, our data clearly show that TULP3 is important for IC3-mediated ciliary targeting. In the discussion, we comment on a few possibilities that might explain these data (**pages 13-14, lines 571-581**).

2. Do the CD8-tagged IC3/CT IC3/CT chimeras bind to Rabl2b? Although the authors show nicely that the IC3 is required in the context of full-length Htr6 for binding to Rabl2, binding in the context of CD8-chimeras would strengthen the idea of direct dependence on Rabl2 binding, particularly for the RKQ motif.

The requested experiment is now in **Fig.12c**. We find that RABL2B binds both CD8-(HTR6-IC3)-myc and CD8-(HTR6-CT)-myc. For HTR6-IC3, we find that mutating its RKQ motif impairs RABL2B binding. In contrast, we see no such effect when disrupting the LPG motif in HTR6-CT. We have updated all relevant sections to reflect this (methods, results, discussion).

Minor comments:

-Fig 9: Data normalized to IPs?

Thanks for raising this point. Although data were already normalized to total TULP3 levels in the IPs, we had omitted to explain that in **Fig.9 and Fig.10 legends**. We have now corrected this.

-Fig 10: Quantification needed. Terms used in describing data such as "moderately reduced", "further reduced" not clear and overtly simplistic without quantification.

We have quantified all Western blot data from **Fig.12** (former Fig.10). Quantifications are displayed at the bottom of all these blots (**Fig.12a-c and Fig.12k**). These data are now more accurately described in the corresponding results sections (**pages 10-11**).

-Fig. 10: "CTS2's positive effect is only unveiled in the absence of HTR6-IC3 residues other than those in CTS1": I am not convinced by the authors' claim of the CTS2's effects without quantification.

As mentioned above, we have quantitated these data and improved their description.

-Fig S1: Should be included among main figures.

Thanks for the suggestion. We have done so. Former Fig.S1 is now **Fig.2**.

-Htr7 is shown to be 20% positive in cilia. (Fig. 2). Probably a fluorescence intensity plot would better substantiate whether the cilia positive for Htr7 are strongly positive or have poor localization, but nonetheless were quantified as "positive".

Great point. Not only is HTR7 in few cilia ($\approx 20\%$) compared to HTR6 ($\approx 100\%$), but ciliary intensity among positive cilia is also about 6-fold higher for HTR6, as we now show in **Fig.S1**.

Reviewer #3 (Comments to the Authors (Required)):

There is accumulating evidence indicating that primary cilia are important signaling hubs which contain many membrane receptors including G protein-coupled receptors (GPCRs). However, the mechanisms underlying ciliary GPCR targeting remain poorly understood, even though ciliary targeting sequences have been identified in a subset of receptors, such as the serotonin 5-HT₆ receptor and the somatostatin SST3 receptor. The paper of Barbeito et al. that explores molecular determinants of ciliary targeting of both receptors, is an important contribution in the field, as it demonstrates that two ciliary targeting sequences contribute to their ciliary location and provide important new insight into the mechanisms by which these sequences regulate binding to TULP3 and RABL2B, two adapters needed for ciliary GPCR targeting. This study also solves the conundrum that the deletion of the previously identified ciliary targeting sequences in the 5-HT₆ and SST3 receptors does not affect their targeting in the primary cilium while they are sufficient to promote ciliary targeting of non-ciliary receptors.

Overall, this is an outstanding study based on a huge number of mutants and chimera, and the paper is very well written. The experiments are well justified and appear to be well done, the data are logically and clearly presented, and the demonstration is generally convincing and easy to follow. Given the neuronal expression of 5-HT₆ and SST3 receptors, the presence of experiments showing the importance of identified ciliary targeting sequences in primary hippocampal neurons certainly gives additional impact to the paper. I have only a few somewhat minor comments that the authors should take into consideration before acceptance of the manuscript.

1- The data on Fig. 1 would be more convincing if additional controls were provided. The authors show a decrease in TULP3 mRNA in siRNA-treated cells and they are right to do that, but they should also demonstrate the efficiency of silencing TULP3 at the protein level. Most importantly, what is the influence of TULP3 knockdown on 5-HT₆ receptor expression? The data on Fig. 1A show that the 5-HT₆ receptor is no more localized in the primary cilium in siRNA-treated cell. However, no expression of the receptor is shown in other compartments. Larger cell fields as well as Western blots showing TULP3 and 5-HT₆ receptor should be represented.

Great point. As requested, we have now analyzed HTR6 and TULP3 by Western blot (**Fig.1d**), and we also show wider cell fields (**Fig.1a**). The Westerns clearly show that: (i) TULP3 protein was successfully silenced, and (ii) HTR6 protein levels are not affected by TULP3 silencing.

Although TULP3 silencing causes loss of ciliary HTR6 staining (**Fig.1a-b**), it does not lead to a detectable increase in extraciliary HTR6 staining. This is a fairly common outcome when ciliary proteins are mislocalized (we saw it repeatedly, for instance, in our previous work: Garcia-Gonzalo et al. 2011 *Nat. Genet.* 43:776). Moreover, this is not too surprising if one considers the dilution effect resulting from moving a given number of HTR6 molecules from a very tiny surface, the ciliary membrane, to a much larger one, the plasma membrane. Our IFs are probably not sensitive enough to detect the small increase at the plasma membrane.

Thus, we think Fig.1 now demonstrates clearly that HTR6 ciliary targeting is TULP3-dependent, and that this is not due to TULP3 affecting HTR6 protein levels. We thank the reviewer for prompting us to make these improvements (and all the ones below).

2- Page 8, bottom: the authors state that both HTR6-CT and SSTR3-CT strongly associated to TULP3, whereas the data on Fig. 8C and 8E show that SSTR3-CT associates much more efficiently to TULP3

than HTR6-CT. Even, HTR6-CT binding to TULP3 does not appear to be significant. Does that mean that 5-HT6 receptor association with TULP3 mainly relies on its targeting sequence located in IC3?

Starting by the last question of the reviewer: our data in **Fig.9b** and **Fig.9d** clearly show no association between HTR6-IC3 and TULP3. Hence, we do not believe HTR6-IC3 is involved in TULP3 association, at least not to an extent that we can detect in our experiments.

Regarding the rest of the comment, we fully agree with the reviewer. We have addressed these points in three ways:

- (1) We no longer say that both HTR6-CT and SSTR3-CT *strongly* associate to TULP3. Instead, in **page 9, lines 358-361**, we now say: “...*both HTR6-CT and SSTR3-CT associated to TULP3 (Fig.9c,e). While SSTR3-CT association was very strong (20-fold over β 2AR-CT control, $p < 0.0001$), HTR6-CT association was less intense (4-fold increase, $p = 0.069$), but still faithfully reproduced in all five independent experiments we performed (Fig.9c,e)*”.
- (2) We repeated this experiment once more. For the fifth time, we saw a clear increase with HTR6-CT. The updated p-value is $p = 0.069$ (down from $p = 0.09$ last time) (**Fig.9e**). Although we are still above $p = 0.05$, we think these data strongly suggest (with 93% confidence) that HTR6-CT specifically interacts with TULP3. In any case, as shown above, we have been very transparent about this in the text, to make sure readers can make their own informed assessments as to the meaning of these data.
- (3) Because of the very high SSTR3-CT values, our previous graph in Fig.9e gave the wrong impression that HTR6-CT levels were very similar to the negative control. We have now addressed this by splitting the y-axis, thereby affording a better view of the data (**Fig.9e**).

3- Page 9 and Fig. 10a: The authors conclude from co-immunoprecipitation of myc-5-HT6 receptor and RABL2B that CTS2 positively regulates the interaction in absence of HTR6-IC3 based on a further reduction of the co-immunoprecipitation when CTS2 is mutated. However, this further reduction is not obvious on the Western blot illustrated on Fig. 10A and the authors should temper this statement in absence on any quantification.

We have quantified all Western blot data from **Fig.12** (former Fig.10). Quantifications are displayed at the bottom of all these blots (**Fig.12a-c** and **Fig.12k**).

Even with the quantitations, we have tempered our statements. The key passage now reads (**page 10, lines 415-420**): “*In contrast, co-IP of myc-Chimera J, lacking HTR6-IC3, and thus also CTS1, was strongly reduced, as was co-IP of myc-Chimera J (CT-mut3), which appeared even lower (Fig.12a) ... This suggests that HTR6-IC3 is important for HTR6-RABL2B binding, and that CTS2 may contribute to the interaction in the context of Chimera J but not of wild type HTR6.*”

4- The profile of 5-HT6 receptor immunoreactivity strongly differs from one blot to one another (see for instance Fig. 10A and S7). In some cases, there is a clear signal at the expected molecular weight for monomer (Fig. S7), whereas the corresponding band seems to be absent in other blots (Fig. 10). How do the authors explain that point?

Great point. What the reviewer says is true. Although the lower molecular weight band ends up appearing in all cases with long enough exposure (**Fig.12a,b,k; Fig.S7; Fig.S10**), the intensity of this band varies across experiments.

The antibody used for blotting may be a factor, as we used anti-HTR6, anti-Flag or anti-Myc in different experiments. SDS-PAGE sample processing may also be a factor. In fact, we have observed

that the lower molecular weight band of HTR6 appears more intensely if samples are treated at 37°C or 65°C, instead of the usual 95°C. However, these low temperatures have their drawbacks, as they interfere with proper denaturation of the other proteins to be analyzed and, in the case of IPs, proteins may not elute well from beads. As a result, we stuck to the standard 95°C protocol for all experiments herein. Still, subtle differences in these high temperature treatments may help explain the observed variation in the low molecular weight band. Other possibilities also exist, but this are our main hypotheses at this point.

To address this issue in the paper, we have explicitly commented on it at the end of the Results section, where we also highlight the uncertainty surrounding this point (**page 11, lines 452-458**).

5- In the introduction, page 2, the authors should quote references when they state that 5-HT6 receptors activate mTOR and Cdk5 signaling. In the same paragraph, regarding the functional outcomes of ciliary targeting of the receptor, they should also quote the paper of Jiang et al. (PNAS 116: 12066-12071, 2019) showing that ciliary localization of 5-HT6 receptor influences constitutive activation of Gs and cAMP production. This is an important recent work in the field.

Thanks for pointing this out. We have now added the requested original references (**page 2, lines 76-81**).

December 11, 2020

RE: Life Science Alliance Manuscript #LSA-2020-00746-TR

Dr. Francesc R Garcia-Gonzalo
Instituto de Investigaciones Biomedicas Alberto Sols UAM-CSIC
Facultad de Medicina UAM (Lab C-11)
Madrid 28029
Spain

Dear Dr. Garcia-Gonzalo,

Thank you for submitting your revised manuscript entitled "HTR6 and SSTR3 ciliary targeting relies on both IC3 loops and C-terminal tails". We would be happy to publish your paper in Life Science Alliance pending final revisions necessary to meet our formatting guidelines.

Along with the points listed below, please also attend to the following:

- please use the [10 author names, et al.] format in your references (i.e. limit the author names to the first 10)
- please add a callout for Figure 4E and Figure S9 in your main manuscript text
- Figure S9 and S10 are source data. please label them as such, instead of labeling them as supplemental figures.

A. FINAL FILES:

-- Summary blurb (enter in submission system): A short text summarizing in a single sentence the study (max. 200 characters including spaces). This text is used in conjunction with the titles of papers, hence should be informative and complementary to the title. It should describe the context

and significance of the findings for a general readership; it should be written in the present tense and refer to the work in the third person. Author names should not be mentioned.

B. MANUSCRIPT ORGANIZATION AND FORMATTING:

Sincerely,

Shachi Bhatt, Ph.D.
Executive Editor
Life Science Alliance
<https://www.lsjournal.org/>
Tweet @SciBhatt @LSAJournal

Reviewer #2 (Comments to the Authors (Required)):

The authors have responded satisfactorily to all my comments. The manuscript should be published without any delay.

A minor comment:

A recent paper in Embo J (PMID:33241915) has described a role of mouse Rab12 in cargo exit rather than entry. I think the authors might consider mentioning this paper in their discussion as well.

Reviewer #3 (Comments to the Authors (Required)):

The authors have made a substantial revision of the manuscript that satisfactorily addressed all my (minor) concerns. This revision strongly improved the manuscript which is an important contribution in the understanding of the mechanisms underlying receptor targeting to the primary cilium.

December 15, 2020

RE: Life Science Alliance Manuscript #LSA-2020-00746-TRR

Dr. Francesc R Garcia-Gonzalo
Instituto de Investigaciones Biomedicas Alberto Sols UAM-CSIC
Facultad de Medicina UAM (Lab C-11)
Madrid 28029
Spain

Dear Dr. Garcia-Gonzalo,

Thank you for submitting your Research Article entitled "HTR6 and SSTR3 ciliary targeting relies on both IC3 loops and C-terminal tails". It is a pleasure to let you know that your manuscript is now accepted for publication in Life Science Alliance. Congratulations on this interesting work.

DISTRIBUTION OF MATERIALS:

Again, congratulations on a very nice paper. I hope you found the review process to be constructive and are pleased with how the manuscript was handled editorially. We look forward to future exciting submissions from your lab.

Sincerely,

Shachi Bhatt, Ph.D.
Executive Editor
Life Science Alliance
<https://www.lsjournal.org/>
Tweet @SciBhatt @LSAJournal